nature
microbiology
## OPEN

# *Wolbachia cifB* induces cytoplasmic incompatibility in the malaria mosquito vector

Kelsey L. Adams[1,2], Daniel G. Abernathy[1,2], Bailey C. Willett [1], Emily K. Selland [1], Maurice A. Itoe[1], and Flaminia Catteruccia [1✉]

***Wolbachia*, a maternally inherited intracellular bacterial species, can manipulate host insect reproduction by cytoplasmic incompatibility (CI), which results in embryo lethality in crosses between infected males and uninfected females. CI is encoded by two prophage genes, *cifA* and *cifB*. *Wolbachia*, coupled with the sterile insect technique, has been used in field trials to control populations of the dengue vector *Aedes albopictus*, but CI-inducing strains are not known to infect the malaria vector *Anopheles gambiae*. Here we show that *cifA* and *cifB* can induce conditional sterility in the malaria vector *An. gambiae*. We used transgenic expression of these *Wolbachia*-derived genes in the *An. gambiae* germline to show that *cifB* is sufficient to cause embryonic lethality and that *cifB*-induced sterility is rescued by *cifA* expression in females. When we co-expressed *cifA* and *cifB* in male mosquitoes, the CI phenotype was attenuated. In female mosquitoes, *cifB* impaired fertility, which was overcome by co-expression of *cifA*. Our findings pave the way towards using CI to control malaria mosquito vectors.***

*W*olbachia endosymbionts are successful insect coloniz-
ers. Some strains of these bacteria induce cytoplasmic
incompatibility (CI) in host insects. CI is the failure of
*Wolbachia*-infected males to produce viable progeny when mated
with uninfected females[1]. Fertility is rescued in females colonized
by *Wolbachia*, providing the endosymbionts with a reproductive
advantage that, when paired with maternal transmission, favours
invasion of insect populations when infection frequencies reach
a certain threshold[2–4]. Two genes (*cifA* and *cifB*) present in WO
prophage regions in the *Wolbachia* genome (with homologues in
all known CI-inducing *Wolbachia* strains) were shown to encode
factors that mediate CI in *Drosophila melanogaster*[5,6]. While it is
known that *cifA* expression in females rescues fertility[7], either one
(*cifB*) or both of these factors are necessary for inducing CI[8–12].

*Wolbachia* biology has attracted considerable interest because
of the potential for exploiting CI in the control of vector-borne
diseases. Control programmes based on the release of *Wolbachia*-
infected mosquitoes to reduce transmission of dengue and other
arboviruses by *Aedes* mosquitoes have advanced to field trials[13–15].
Additionally, a strategy known as Incompatible Insect Technique
(IIT), which uses the infertility induced by *Wolbachia*-infected
males mating with uninfected females to achieve suppression of
insect populations, has been successfully applied in field trials of
*Aedes* mosquitoes[16,17].

The implementation of similar *Wolbachia*-based strategies to
tackle malaria-transmitting *Anopheles* mosquitoes holds appeal
because widespread insecticide resistance threatens strategies for
vector control[18,19]. However, *Wolbachia* does not appear to form
stable endosymbiosis with *Anopheles* species. Although there is evi-
dence that *Wolbachia* can limit the ability of *Plasmodium* to infect
*Anopheles* mosquitoes[20–25], only one artificial *Wolbachia* infec-
tion has been achieved in the germline of an anopheline species.
Moreover, upon endosymbiosis of *An. stephensi* with a *w*AlbB
strain (a strong CI-inducing strain from *Aedes albopictus*), only
partial rescue of CI and limited capacity for population invasion
were observed[26]. With the exception of one report of high-density

*Wolbachia* infection in *An. moucheti* and *An. demeilloni*[27], only few
low-titre natural *Wolbachia* infections have been reported in field
populations of *Anopheles*[23,28–32], and these findings have been ques-
tioned by some researchers[33].

We hypothesized that *cifA* and *cifB* genes alone might be capable
of inducing CI in *An. gambiae*, the most important malaria vector
in Africa, and report our findings here.

## Results

***cifA* and *cifB* expression in *An. gambiae* induces embryonic
lethality.** We chose to use the Type I *cif* genes, *cifA* and *cifB*, from
the *Wolbachia* strain *w*Pip, which induces strong CI in its natural
mosquito host, *Culex pipiens*. Two different nomenclature systems
exist for CI in *Wolbachia*, and the *w*Pip Type I *cif* genes are also
known as CI-inducing deubiquitinases (*cidA* and *cidB*)[6,34]. After
codon optimization, we separately cloned each *w*Pip gene under the
control of the *zero population growth (zpg)* promoter[35], which drives
germline-limited expression in both male and female germ cells of
*An. gambiae*[36] (Fig. 1a). Co-injection of *zpg-cifA* and *zpg-cifB* con-
structs yielded F1 transgenics expressing either *cifA* alone, or both
*cifA* and *cifB* (*zpg-cifA;B*), but none that expressed *cifB* only.

We set up crosses between *zpg-cifA;B* males and different female
lines (*zpg-cifA;B*, *zpg-cifA* and wild type (WT) females), using WT
males as control. In all crosses, females mated to *zpg-cifA;B* males
showed a striking degree of infertility (only 2–4% viable progeny)
compared with controls (Fig. 1b). Most infertile embryos were
arrested early in development, but a minority initiated develop-
ment then failed to hatch (Extended Data Fig. 1). Embryo cytology
revealed the hallmarks of CI[5,6,37], with most embryos showing early
developmental arrest, while others showed fewer nuclear divisions
or were arrested later in the blastoderm stage due to mitotic fail-
ures (Fig. 1c). We did not observe any substantive rescue of infertil-
ity when females expressed either *cifA* alone, or both *cifA* and *cifB*
(Fig. 1b). We also observed a minor (17%) decrease in fertility of
*zpg-cifA;B* females compared with their WT and *zpg-cifA* counter-
parts when mated with WT males (Fig. 1b).

[1]Department of Immunology and Infectious Diseases, Harvard T.H. Chan School of Public Health, Boston, MA, USA. [2]These authors contributed equally:
Kelsey L. Adams, Daniel G. Abernathy. ✉e-mail: fcatter@hsph.harvard.edu

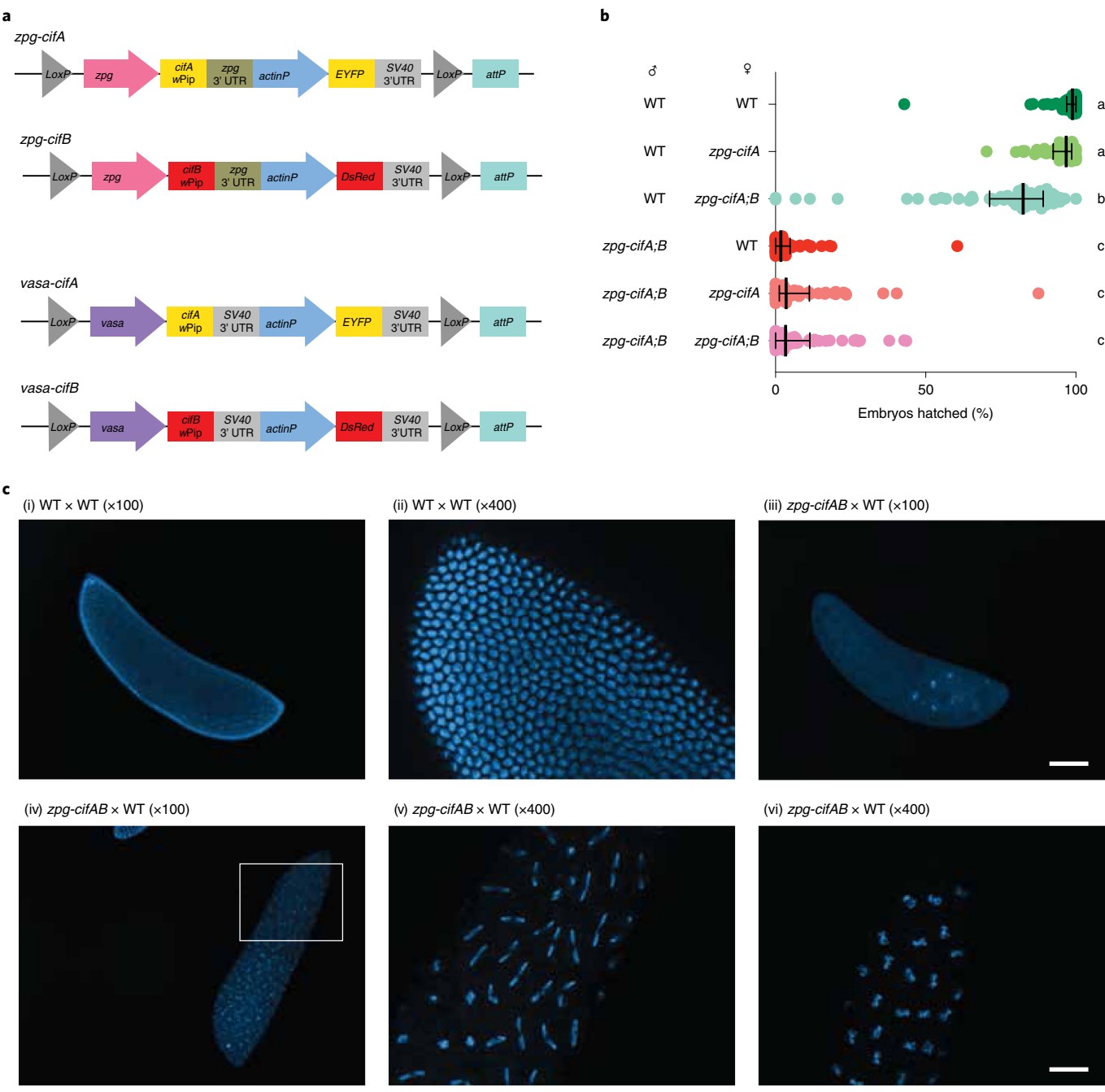

**Fig. 1 | Co-expression of *cifA* and *cifB* in male *An. gambiae* causes embryonic lethality in progeny. a**, Construct design of *zpg-cifA*, *zpg-cifB, vasa-cifA* and *vasa-cifB*. **b**, Males that express *zpg-cifA;B* produced largely inviable progeny, regardless of whether their female mate expresses *zpg-cifA*. Expression of *zpg-cifA;B* in females caused a decrease in female fertility compared with WT females, but expression of *cifA* alone did not (Dunn's multiple comparisons test (two-sided), $P \leq 0.0071$ for groups a vs b, $P < 0.0001$ for groups a vs c and groups b vs c). Median and interquartile ranges are shown. For each group (top to bottom), the *n* (number of broods) is as follows: 58, 52, 59, 51, 53, 62. Kruskal–Wallis test: $H = 265$, $P < 0.0001$, d.f. = 5. **c**, Embryos from *zpg-cifA;B* males crossed with WT females (or WT crosses, as controls in (i) and (ii)) were fixed and imaged with DAPI 3–4 h post oviposition, showing developmental arrest of most CI embryos during early nuclear divisions (iii), while some embryos completed multiple rounds of nuclear division but showed mitotic defects, such as chromatin bridging ((iv), with a close-up in (v)), and other chromosomal abnormalities resulting in delayed or arrested development (vi). Scale bars, 100 μm for ×100 images and 25 μm for ×400 images.

**High expression levels of *cifA* in females rescues CI in *An. gambiae*.** We speculated that the lack of fertility rescue by *zpg-cifA* could be due to insufficient expression of *cifA* in females, as the rescue effect has been shown to be promoter-dependent[7]. To test this possibility, we engineered transgenic expression of *cifA* from the *vasa* promoter[38] (Fig. 1a) because the *vasa* promoter has considerably

higher expression levels in the female germline than the *zpg* promoter (Fig. 2a). In *D. melanogaster*, in addition to different expression levels, *vasa* has a different expression pattern compared with *zpg* as it is also expressed in somatic gonadal precursors[39], although it is not known whether this is true in *Anopheles*[38]. When mated to *zpg-cifA;B* males, high levels of infertility were observed in both

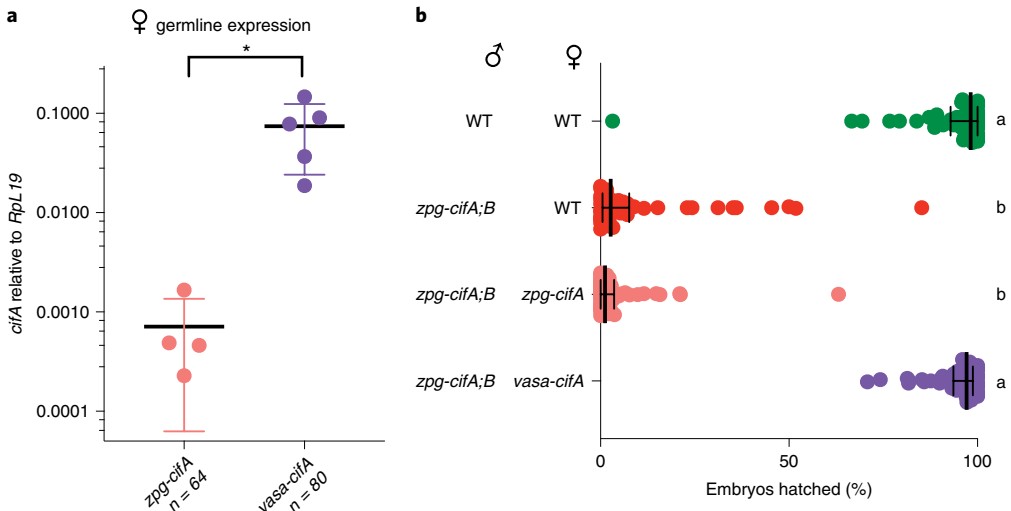

**Fig. 2 | High expression of female *cifA* rescues *cifA;B*-induced CI in *An. gambiae*. a**, Transcript abundance of *cifA* was higher in *vasa-cifA* females compared with *zpg-cifA* females relative to *RpL*19 (unpaired *t*-test (two-tailed), *P* = 0.0232, mean and s.d. are shown). **b**, The expression of *vasa-cifA* in females rescued infertility caused by *zpg-cifA;B* expression in males to WT levels, while expression of *zpg-cifA* in females did not (Dunn's multiple comparisons test (two-sided), *P* < 0.0001 for differences between all statistical groups). Median and interquartile ranges are shown. For each group (top to bottom), the *n* is as follows: 51, 50, 52, 52. Kruskal–Wallis test: H = 153.1, *P* < 0.0001, d.f. = 3.

*zpg-cifA* and WT females as above, but in this case fertility was fully restored in crosses with *vasa-cifA* females, demonstrating effective rescue by this transgene (Fig. 2b). Combined, these results reveal that CI can be recapitulated in *An. gambiae* mosquitoes by transgenic expression of *cifA* and *cifB*. We also attempted co-injections of *vasa-cifA* and *vasa-cifB* constructs (Fig. 1a), but failed to isolate any *cifB*-expressing progeny.

**Expression of *cifB* alone in males induces CI.** Next, we investigated whether *cifB* alone can induce CI. Although we could not maintain a *zpg-cifB* colony in the absence of *cifA*, we were able to isolate, by fluorescent screening, a limited number of F1 *zpg-cifB* males from natural colony matings between heterozygous mosquitoes. We found that *zpg-cifB* males induced high infertility when mated to WT females, at a rate that was statistically indistinguishable from the infertility levels induced by *zpg-cifA;B* males (Fig. 3a). In contrast, we found that progeny sired by *zpg-cifA* males were fully fertile (Fig. 3a). CI induction did not differ whether *zpg-cifB* males were isolated from *zpg-cifA;B* or *vasa-cifA;zpg-cifB* colonies (denoted (*z*)*zpg-cifB* or (*v*)*zpg-cifB*, respectively) (Fig. 3b). We also showed that *vasa-cifA* expression in females was sufficient to completely rescue sterility caused by *zpg-cifB* males, ruling out CI-independent effects (Fig. 3b). Cytology of 69 embryos confirmed the results obtained with *zpg-cifA;B* males, revealing the canonical features of CI (Extended Data Fig. 2). These findings show that conditional sterility can be induced by *cifB* alone in mosquitos.

**cifA expression at high levels in males attenuates CI.** Given that *vasa-cifA* rescues inviability caused by *cifB* in the embryo while *zpg-cifA* does not, we next asked whether expressing *cifA* under the *vasa* promoter in males may impact the strength of CI. To this end, we compared fertility of crosses between males expressing either *zpg-cifA;B* or *vasa-cifA;zpg-cifB* and WT females (Fig. 3c). Intriguingly, *vasa-cifA;zpg-cifB* males were considerably more fertile (median of 48% hatched embryos) compared with *zpg-cifA;B* males (median of 0% hatched embryos) (Fig. 3c). Consistent with female data, *cifA* expression was higher in *vasa-cifA;zpg-cifB* males compared with *zpg-cifA;zpg-cifB* males, while *cifB* expression levels were similar (Fig. 3d). Also in this case, the intermediate sterility effects

caused by *vasa-cifA;zpg-cifB* males were rescued when females expressed *vasa-cifA* (Extended Data Fig. 3). Further, *vasa-cifA* expression in males did not induce embryonic lethality, supporting the idea that *vasa-cifA* does not contribute to CI in males and acts solely as a rescue factor (Extended Data Fig. 3). Higher expression of *cifA* in males (and/or a difference in localization or timing of expression compared with those achieved by the *zpg* promoter) thus reduces CI penetrance rather than favoring it, possibly either by limiting CifB activity within the male germline, or by rescuing CifB toxicity in the embryo following transfer of CifA in sperm[40].

**cifB expression in females disrupts fertility and fecundity.** Our finding that *cifB* expression in males is sufficient to induce significant sterility prompted us to investigate toxicity of this factor in females. We designed crosses between WT males and either *zpg-cifA;B* or *vasa-cifA;zpg-cifB* females (Fig. 4a) and then characterized egg development and fertility of the *zpg-cifB* F1 female progeny after mating to WT males. We noticed that, in contrast to males (Fig. 3b), *cifB*-mediated effects in females were dependent on the colony of origin. When derived from *zpg-cifA;B* mothers, most F1 *zpg-cifB* females (called (*z*^mat)*zpg-cifB*) failed to develop eggs following a blood meal, and only a few females yielded fertile progeny (Fig. 4b,c). Additionally, morphological analysis of the ovaries before and after ingestion of a blood meal showed that follicles were largely absent, suggestive of defects in germline development (Fig. 4d,e). When derived from *vasa-cifA;zpg-cifB* mothers, F1 females ((*v*^mat)*zpg-cifB*) showed intermediate phenotypes with substantial follicle development, although both fecundity and fertility were reduced compared with WT females (Fig. 4b–e). However, when the *cifB* transgene was inherited from *vasa-cifA;zpg-cifB* fathers (Fig. 4a), most F1 females ((*v*^pat)*zpg-cifB*) had ovaries similar to those of (*z*^mat)*zpg-cifB* females, showing remarkably reduced follicle development (Fig. 4d,e). As the *zpg-cifB* insertion site and promoter is the same in all these groups, these results reveal rescue effects possibly caused by maternal deposition of *cifA* (as either mRNA of protein) from *vasa-cifA*-expressing mothers, although we cannot rule out a difference in other host factors. This is consistent with data showing that transgenes expressed under the *vasa* promoter, but not the *zpg* promoter, are maternally deposited[38,41].

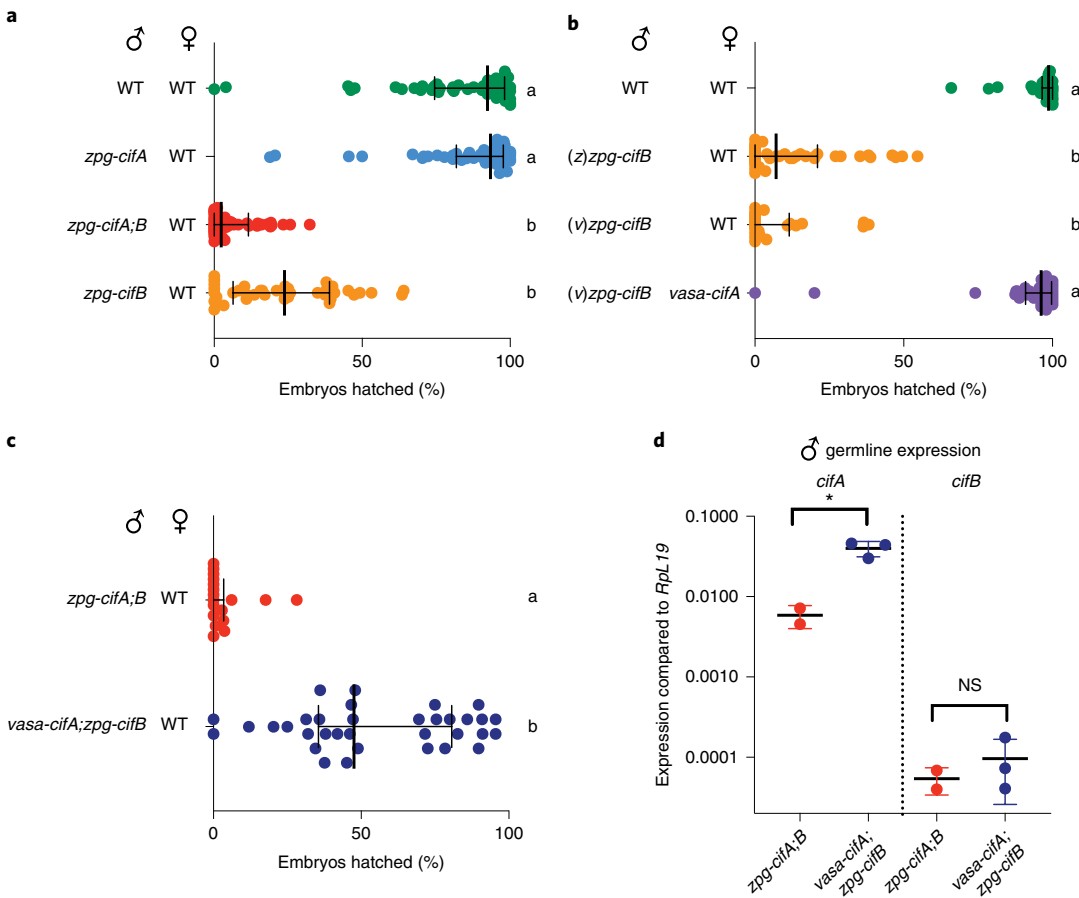

**Fig. 3 | Male *cifB* expression is sufficient to cause CI, while male *cifA* attenuates it. a**, *zpg-cifB* males caused infertility in WT females, while *zpg-cifA* males did not (Dunn's multiple comparisons test (two-sided), $P \leq 0.0001$ for differences between all statistical groups). Median and interquartile ranges are shown. For each group (top to bottom), the *n* is as follows: 55, 55, 44, 39. Kruskal–Wallis test: $H = 133.8$, $P < 0.0001$, d.f. = 3. **b**, The expression of *vasa-cifA* in females rescued infertility caused by (*v*)*zpg-cifB* expression in males, which induced CI to the same extent as (*z*)*zpg-cifB* males (Dunn's multiple comparisons test (two-sided), $P \leq 0.0001$ for differences between all statistical groups). Median and interquartile ranges are shown. For each group (top to bottom), the *n* is as follows: 36, 39, 24, 32. Kruskal–Wallis test: $H = 95.08$, $P < 0.0001$, d.f. = 3. **c**, Expression of *vasa-cifA;zpg-cifB* in males caused only partial induction of CI (Mann–Whitney test (two-tailed), $P < 0.0001$). Median and interquartile ranges are shown. For each group (top to bottom), the *n* is as follows: 18, 34. **d**, Expression of *cifA* in the male germline was higher in *vasa-cifA* than in *zpg-cifA* (unpaired *t*-test (two-tailed), *$P = 0.0135$, mean and s.d. are shown), while the expression of *cifB* was similar (unpaired *t*-test (two-tailed), $P = 0.4882$, mean and s.d. are shown). For each group (left to right), the total *n* is as follows: 32, 48, 32, 48.

*cifB* expression is therefore highly deleterious in the female germline when unchecked by the presence of *cifA*, and it seems to function during the early stages in germline development based on the capacity for maternally derived *cifA* to rescue these defects.

## Discussion

Using *cif* genes from *w*Pip in *An. gambiae*, we show that *cifB* expression is sufficient to induce embryonic lethality via CI. We show that it is possible to induce and rescue CI in *An. gambiae*, suggesting that it may be feasible to apply *Wolbachia* or *Wolbachia*-derived genes for anopheline vector control[13]. However, the reproductive toxicity observed in both sexes upon *cifB* expression may partially explain why infections using CI-inducing *Wolbachia* strains have been difficult to establish in laboratory colonies of these mosquitoes. Previous efforts to generate *cifB^wPip*-expressing *D. melanogaster* were unsuccessful[6], consistent with our own difficulties in isolating *cifB*-expressing individuals using two different promoters unless *cifA* was also co-expressed, and with our results demonstrating *cifB* toxicity.

Our findings are in contrast with results reported for CI in *D. melanogaster* where both *cifA* and *cifB* from *w*Mel were required

to induce CI, and where a *cifB* transgenic line was isolated in the absence of *cifA*[5]. Further, *cifB^wMel* females showed no defects in fertility, contrary to our results[5]. Many possible reasons could explain these discrepancies, ranging from different promoters and transgene insertion sites to specific differences in CifB function in its natural host (such as in the case of *cifB* from *w*Mel in *D. melanogaster*) compared to a novel host (*cifB* from *w*Pip in *An. gambiae*). Additionally, the observation that *w*Mel causes weak CI in its natural host *D. melanogaster*[42,43] (though it induces strong CI in *Drosophila simulans* and *Ae. aegypti*[44,45]), while *w*Pip causes strong CI in its natural host *C. pipiens*[46], highlights the possibility for host-dependent and strain-dependent differences. In future studies, it will be interesting to determine whether *w*Mel *cifB* can induce embryonic lethality in *An. gambiae*.

Other studies have shown infertility induced by *cifB* alone in *D. melanogaster*, induced by *w*Pip's Type IV *cifB* homologue (also called *cinB*) and by the Type I *cifB* homologue from *w*Rec, a CI-inducing strain found in *Drosophila recens*[47,48]. Neither study was able to demonstrate rescue of these effects and thus could not conclude that they were CI related[47,48]. However, when *cifA* was

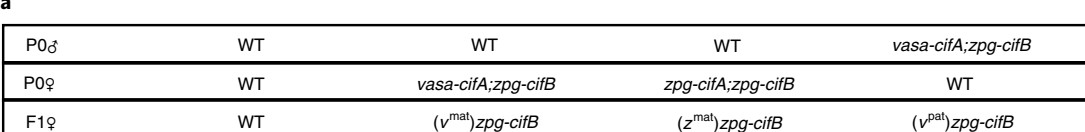

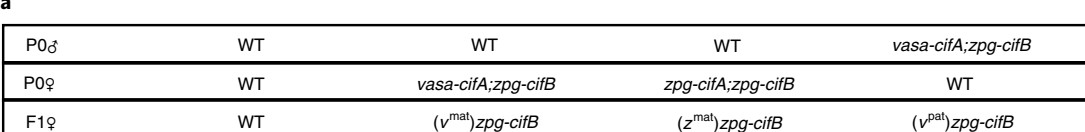

**Fig. 4 | cifB expression in females causes severely impaired follicle development in the absence of cifA. a**, Crosses were set up to isolate *zpg-cifB* females, F1 progeny derived from either mothers that also expressed *vasa-cifA* ((*v*ᵐᵃᵗ)*zpg-cifB*) or *zpg-cifA* ((*z*ᵐᵃᵗ)*zpg-cifB*), or fathers also expressing *vasa-cifA* ((*v*ᵖᵃᵗ)*zpg-cifB*). **b**, Egg development was nearly abolished in (*z*ᵐᵃᵗ)*zpg-cifB*-expressing females, while nearly all (*v*ᵐᵃᵗ)*zpg-cifB* females showed egg development, although with decreased numbers of eggs compared with WT females (Dunn's multiple comparisons test (two-sided), $P < 0.0001$ for differences between all statistical groups). Medians and interquartile ranges are shown. For each group (top to bottom), the *n* is as follows: 68, 66, 68. Kruskal–Wallis test: $H = 153.9$, $P < 0.0001$, d.f. = 2. **c**, (*z*ᵐᵃᵗ)*zpg-cifB* and (*v*ᵐᵃᵗ)*zpg-cifB* females showed impaired fertility compared with WT females (Dunn's multiple comparisons test (two-sided), $P < 0.0257$ between groups b and c, $P < 0.0001$ for other comparisons). Median and interquartile ranges are shown. For each group (top to bottom), the *n* is as follows: 60, 54, 17. Kruskal–Wallis test: $H = 64,64$, $P < 0.0001$, d.f. = 2. **d,e**, Ovaries from *cifB* females showed severely impaired follicle development unless derived from a *vasa-cifA*-expressing mother, when imaged at either 0 h or 24 h post blood feeding (p.b.f.) before fixing under brightfield microscopy (**d**) (scale bar, 800 µm) or after fixing using differential interference contrast microscopy (**e**) (scale bar, 100 µm).

expressed alongside *cifB*ʷᴿᵉᶜ in males, very little embryonic lethality was observed, reminiscent of our results showing that high *cifA* expression in males can attenuate CI[47].

Combined with the data we present indicating that high levels of *cifA* can rescue CI in females but attenuate *cifB* activity in males, it is possible that *Wolbachia* may need to fine-tune the relative expression

of *cifA* and *cifB* in males and females to induce CI in *Anopheles* mosquitoes without causing lethal toxicity. Such a balancing act might make maintenance of CI-inducing *Wolbachia* strains in anopheline insects difficult and one outcome could be silencing of the toxic *cifB* gene by mutation. Interestingly, *cifB* nonsense mutations were identified by sequencing *w*AnM and *w*AnD strains recently discovered in some *Anopheles* species, although one *cifB* homologue in *w*AnD seems to be intact and it will be interesting to learn if this strain can induce CI[27]. Of note, *cifB* pseudogenization is not uncommon and is consistent with early evolutionary models that predict male incompatibility not to be selected for within a host lineage[49,50].

It may be possible to enable stable colonization of *Anopheles* by *Wolbachia* by limiting *cifB* toxicity using germline expression of *cifA*. This would create a route to screening for *Wolbachia* strains that can block transmission of *Plasmodium* parasites and pave the way to using *Wolbachia* endosymbiosis in population replacement strategies for malaria control. Plus, the remarkable sterility induced by *cifB* or *cifA;B* co-expression could be utilized for sterile male releases to suppress *Anopheles* populations even in the absence of *Wolbachia* infection, similar to the IIT programmes implemented in *Aedes* mosquito control[16,17]. Due to the difficulty in rearing both *cifB* and *cifA;B* mosquitoes, conditional expression of these genes may be required if mass releases were to occur. At a time when novel malaria control strategies are urgently needed, our data presents a step towards utilizing *Wolbachia, or Wolbachia*-derived genes, in control programmes targeting *Anopheles* mosquitoes.

## Methods

**Generation of constructs.** The amino acid sequences for *cifA* (wPa_0282) and *cifB* (wPa_0283) coding regions from the published *w*PipI Pel strain of *w*Pip from *C. pipiens*[51] were codon-optimized for expression in *An. gambiae* using published codon bias information[52]. Gene blocks were ordered from Integrated DNA Technologies, using custom gene synthesis to create the desired DNA fragments (accession codes OK352257 (*cifA*) and OK352258 (*cifB*)). Transgenesis constructs were engineered to express the *w*Pip CI genes *cifA* and *cifB* under the control of the germline-specific promoters *zpg* (*zpg*, AGAP006241)[35] and *vasa* (*vasa2*, AGAP008578)[38,53]. The constructs also express a fluorescent marker under control of the ubiquitous *actin* promoter to enable selection of transgenic mosquitoes. Integration into the mosquito genome was mediated by *piggyBac* transposition and rearing lines to homozygosity was accomplished through pupae sorting via fluorescence intensity.

**Embryonic micro-injection.** PiggyBac transgenic construct pairs corresponding to each germline promoter (*zpg-cifA-EYFP* and *zpg-cifB-DsRed*; or *vasa-cifA-EYFP* and *vasa-cifB-DsRed*) were co-injected into the posterior of freshly laid embryos from *An. gambiae* (*vasa* n = 1,434, *zpg* n = 512) at a concentration of 250 ng μl⁻¹. Pupae that survived injection were separated according to sex, reared to adulthood, and backcrossed to wild-type G3 to identify and isolate transgenics. A total of 17 EYFP/DsRed double positive F1 transgenics were recovered from the *zpg* promoter-driven CI constructs injections. In contrast, only *vasa-cifA-EYFP* positive transgenics were recovered from the *vasa-cifA/cifB* co-injections. Irrespective of germline promoter, no *cifB* transgenic mosquitoes were identified post-injection.

**Mosquito lines and rearing.** *An. gambiae* mosquitoes from the G3 strain and transgenic derivatives of the G3 strain were maintained in a 27 °C insectary environment with 70–80% humidity and a 12 h light:12 h dark cycle. Adults were given 10% glucose and water ad libitum and fed on human blood (Research Blood Components). Larvae were fed a mixture of Tetramin fish flakes and pellets.

Separate colonies containing the following transgenes were maintained: Colony 1, *zpg-cifA* (Chromosome 3R insertion); Colony 2, *zpg-cifA* (3R); *zpg-cifB* (Chromosome X insertion); Colony 3, *vasa-cifA* (putative Chromosome 2L insertion (2L*)); Colony 4, *vasa-cifA* (2L*); *zpg-cifB* (X); and Colony 5, *zpg-cifA* (unknown insertion); *zpg-cifB* (X). To establish Colony 4, *zpg-cifB* males isolated from Colony 2 or 5 were crossed with *vasa-cifA* females from Colony 3. All colonies were maintained as heterozygotes and screened for fluorescent markers as pupae to select for the presence of transgenes. For all experiments using *zpg-cifA* mosquitoes, Colony 1 was used. For all experiments using *zpg-cifA;B* mosquitoes, Colony 2 was used. For all experiments using *vasa-cifA* mosquitoes, Colony 4 was used. Experiments using (*z*)*zpg-cifB* mosquitoes used mosquitoes isolated from either Colony 2 or 5. Experiments using (*v*)*zpg-cifB* males used mosquitoes isolated from Colony 4.

**Crosses and fertility assays.** To perform crosses between different transgenic lines, individuals were isolated as pupae from these colonies and their transgenes were identified by their respective fluorescent markers. We did not verify whether individuals were homozygous or heterozygous for their transgenes. Pupae were separated by sex under a dissecting microscope, placed in cages with a male to female ratio between 1:1 and 2:1, and allowed to eclose in small BugDorm cages. Natural mating proceeded and mosquitoes were given ad libitum access to 10% glucose solution and water for 5–7 d before blood-feeding females and allowing oviposition in individual cups lined with filter paper. Once laid, eggs were stimulated daily by spraying water and allowed to hatch for a minimum of 4 d. We then assessed fertility of females by counting and scoring eggs under a Leica M80 dissecting microscope, and additionally noting the presence or absence of hatched larvae. For any female that showed no fertile embryos, mating status was verified by checking microscopically for the presence of sperm in the spermatheca. For egg development experiments, egg counts for all females were included regardless of whether they had mated or oviposited, while only those that were mated and oviposited were included in fertility experiments.

**RNA extraction and quantitative reverse transcription PCR (RT–qPCR).** Male or female reproductive tracts were dissected in pools of 16, collected in TRI reagent (Thermo Fisher Scientific), and stored at −80 °C. RNA was extracted according to the manufacturer's instructions, with an additional three ethanol washes of pelleted RNA. Following resuspension, RNA was treated with Turbo DNAse (Thermo Fisher Scientific), quantified with a Nanodrop 2000C (Thermo Fisher Scientific), and then 0.75–2 μg were used in a 100 μl complementary DNA synthesis reaction, following standard protocols. We designed primers for RT–qPCR (QuantStudio 6 pro, Thermo Fisher Scientific) using NCBI PrimerBLAST[54] and after evaluating four different primer sets for *cifB*, we used the following primers for *cifA* and *cifB* at the following concentrations: cifAF, 5′ tcgccgagctgatcgtgaa 3′ (300 nM); cifAR, 5′ atcatgtccaggatctccttcttctc 3′ (300 nM); cifBF, 5′ AGAAGGACCGCCTGATCG 3′ (900 nM); cifBR, 5′ AGGCTATCGGCGTAGTAGCC 3′ (900 nM); RpL19F, 5′ CCA ACTCGCGACAAAACATTC 3′ (300 nM); and RpL19R, 5′ ACCGGCTTCTTGA TGATCAGA 3′ (900 nM). Relative quantification was determined using the $2^{-(\Delta Ct \text{ (cycle threshold)})}$ equation, using *RpL19* as the standard. For female *cifA* expression (Fig. 2a), transcript levels were not found to be different between samples from *cifA* only or *cifA* and *cifB* co-expressing individuals, so these data were pooled.

**Microscopy and tissue staining.** *Embryo cytology.* Embryos were collected from 10–12 WT females after natural matings with *zpg-cifA;B* males. Four hours after oviposition, embryos were bleached, washed and dechorionated according to methods by Goltsev et al.[55], and the endochorion was peeled according to methods by Juhn and James[56]. Embryos were then fixed and stained with 4′,6-diaminidino-2-phenylindole and imaged on a Zeiss Inverted Observer Z1 at ×100 or ×400 magnification.

*Brightfield microscopy of embryos.* A sample of oviposited embryos were imaged on filter paper at either ×5 or ×7.5 on a Leica M80 dissecting scope.

*Brightfield and differential interference contrast imaging of ovaries.* Ovaries of 4–7-day-old females were dissected in PBS at either 0 h or 24 h post-blood-meal and imaged with a Leica M80 dissecting scope at ×7.5 magnification for general morphology. After initial imaging, ovaries were fixed in 4% paraformaldehyde and then mounted in Vectashield mounting medium with 4′,6-diaminidino-2-phenylindole counterstain (Vector Laboratories). Ovaries were then imaged using differential interference contrast on a Zeiss Inverted Observer Z1 at ×100 magnification.

**Statistical methods.** In all comparisons of fertility or egg development, Anderson–Darling normality tests showed that all data were not normally distributed, so non-parametric Kruskal–Wallis tests with Dunn's multiple comparisons were used. Distinct samples were used for comparisons. Tests were performed in GraphPad Prism 8. For all fertility or egg development experiments, 2–4 replicates were performed for all groups. We conducted power analysis in G*Power 3.1 to detect a 50% reduction in fertility, yielding n = 5, non-centrality parameter = 21.3, critical chi-squared = 11.07 and total sample size = 10 to estimate the sample size required for detecting differences. On average, we used far greater sample sizes than this power analysis suggested, as we planned to use more stringent tests for non-parametric data (which cannot be estimated by power analysis). Only one replicate was performed for embryo cytology experiments. For Fig. 4d,e, where representative images were selected, one replicate of dissection and imaging of 5–10 individuals from each group was performed; however, these phenotypes were confirmed by dissections performed in experiments for Fig. 4b,c. For all RT–qPCR data, 2–5 technical replicates with 16 individuals each were performed for each group, with exact n given in figure legends.

**Reporting Summary.** Further information on research design is available in the Nature Research Reporting Summary linked to this article.

## Data availability

Source data are provided with this paper. Sequence information can be found in GenBank with the accession numbers OK352257 (codon-optimized *cifA*) and OK352258 (codon-optimized *cifB*).

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

## Acknowledgements

We thank W. R. Shaw for careful reading of the manuscript and advice, and A. Smidler for help with construct design. This study was funded by a joint Howard Hughes Medical Institute (HHMI) and Bill and Melinda Gates Foundation (BMGF) Faculty Scholars Award to F.C. (Grant ID: OPP1158190), a BMGF research grant (Grant ID: OPP1174120) and by a fellowship of the National Sciences and Engineering and Research Council of Canada (NSERC) to K.A. The findings and conclusions in this publication are those of

the authors and do not necessarily reflect positions or policies of the HHMI, the BMGF or the NSERC.

## Author contributions

K.L.A. contributed to literature searches, study design, data collection, data analysis, data interpretation, figure creation and writing; D.G.A. contributed to literature searches, study design, data collection, data analysis, data interpretation and writing; B.C.W. contributed to literature searches, study design, data analysis and data collection; E.K.S. contributed to data analysis and data collection; M.A.I. contributed to data collection; F.C. supervised the project and contributed to study design, data analysis, data interpretation, figure creation and writing. The corresponding authors had full access to all data in the study and final responsibility in the decision to submit for publication.

## Competing interests

The authors declare no competing interests.

## Additional information

**Extended data** is available for this paper at https://doi.org/10.1038/s41564-021-00998-6.

**Correspondence and requests for materials** should be addressed to Flaminia Catteruccia.

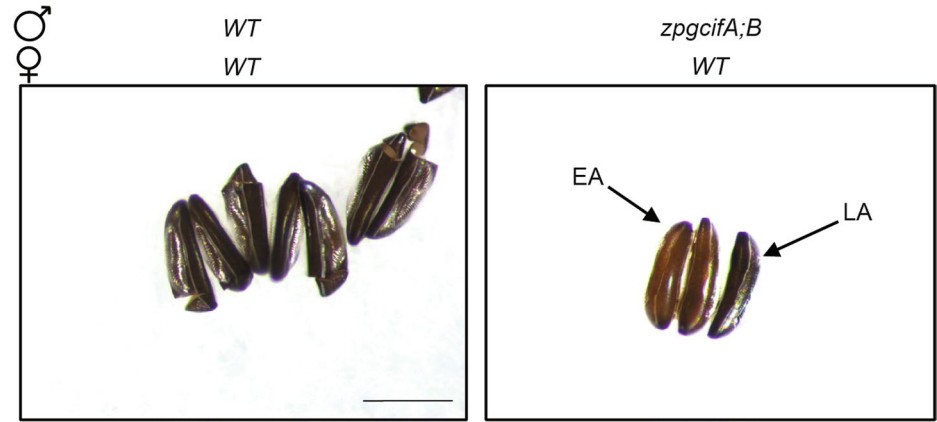

**Extended Data Fig. 1 | Embryos from *zpg-cifA;B* males show either early or late arrest.** Brightfield images of eggs 5 days after oviposition from crosses between WT mosquitoes (left) or *zpg-cifA;B* males and WT females (right). While WT embryos show full development and the standard opening of the hatching cap following larval hatching, embryos from *zpg-cifA;B* males are inviable and arrested either during early development (EA) with a pale brown color, or late development (LA), which show stemmata, but also present severe abnormalities and do not hatch. Scale bar represents 400μm.

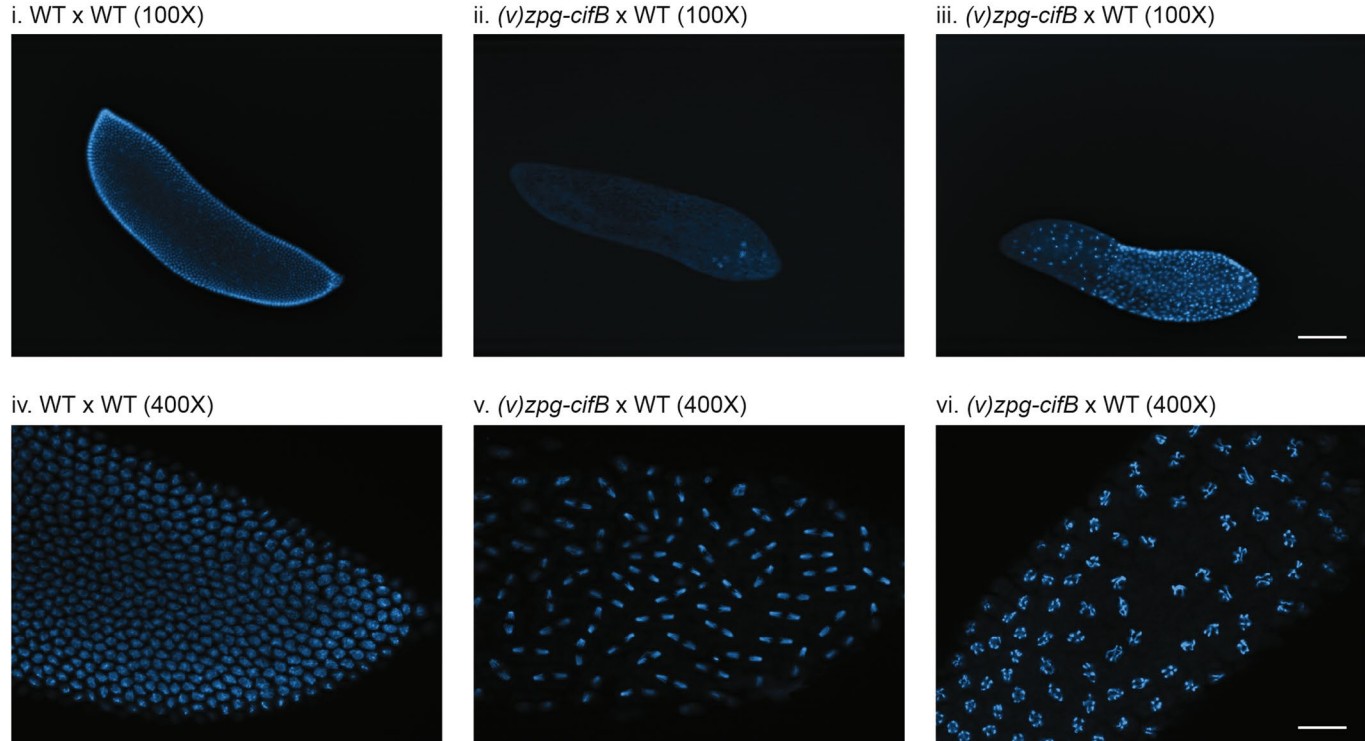

i. WT x WT (100X)  ii. *(v)zpg-cifB* x WT (100X)  iii. *(v)zpg-cifB* x WT (100X)

iv. WT x WT (400X)  v. *(v)zpg-cifB* x WT (400X)  vi. *(v)zpg-cifB* x WT (400X)

**Extended Data Fig. 2 | Cytology of the progeny of *cifB* males shows hallmarks of CI.** F1 embryos of crosses between either WT or *(v)zpg-cifB* males with WT females were stained for DAPI and imaged 3-4 hours after oviposition. At 100X, while WT controls (i. and iv.) show normal development, *(v)zpg-cifB* embryos show various hallmarks of CI, including (ii.) early arrest, (iii.) regional mitotic failure, and (v., vi.) chromatin bridging or chromosomal abnormalities and delayed or arrested nuclear division. Scale bars represent 100μm for 100X images and 25μm for 400X images.

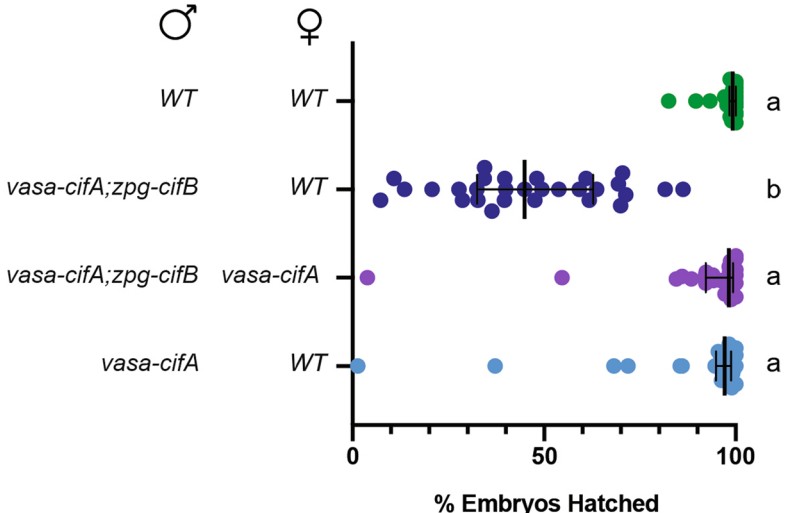

**Extended Data Fig. 3 | *vasa-cifA* does not cause CI, and likely inhibits its induction when co-expressed in males.** *vasa-cifA* expression alone in males does not cause CI, and its expression in females is sufficient to rescue the intermediate CI phenotype caused by expression of *vasa-cifA;zpg-cifB* in males. (Dunn's multiple comparisons (two-sided), $p \leq 0.0001$ for differences between all statistical groups). Median and interquartile ranges are shown. For each group (top to bottom) the *n* is as follows: 30, 29, 28, 28. Kruskal-Wallis results: H = 62.87, $p < 0.0001$, df=3.

# Reporting Summary

Nature Research wishes to improve the reproducibility of the work that we publish. This form provides structure for consistency and transparency in reporting. For further information on Nature Research policies, see our Editorial Policies and the Editorial Policy Checklist.

## Statistics

For all statistical analyses, confirm that the following items are present in the figure legend, table legend, main text, or Methods section.

| n/a | Confirmed | |
|---|---|---|
| ☐ | ☒ | The exact sample size (*n*) for each experimental group/condition, given as a discrete number and unit of measurement |
| ☐ | ☒ | A statement on whether measurements were taken from distinct samples or whether the same sample was measured repeatedly |
| ☐ | ☒ | The statistical test(s) used AND whether they are one- or two-sided *Only common tests should be described solely by name; describe more complex techniques in the Methods section.* |
| ☐ | ☒ | A description of all covariates tested |
| ☐ | ☒ | A description of any assumptions or corrections, such as tests of normality and adjustment for multiple comparisons |
| ☐ | ☒ | A full description of the statistical parameters including central tendency (e.g. means) or other basic estimates (e.g. regression coefficient) AND variation (e.g. standard deviation) or associated estimates of uncertainty (e.g. confidence intervals) |
| ☐ | ☒ | For null hypothesis testing, the test statistic (e.g. *F*, *t*, *r*) with confidence intervals, effect sizes, degrees of freedom and *P* value noted *Give P values as exact values whenever suitable.* |
| ☒ | ☐ | For Bayesian analysis, information on the choice of priors and Markov chain Monte Carlo settings |
| ☒ | ☐ | For hierarchical and complex designs, identification of the appropriate level for tests and full reporting of outcomes |
| ☒ | ☐ | Estimates of effect sizes (e.g. Cohen's *d*, Pearson's *r*), indicating how they were calculated |

*Our web collection on statistics for biologists contains articles on many of the points above.*

## Software and code

Policy information about availability of computer code

| Data collection | N/A |
|---|---|
| Data analysis | N/A |

For manuscripts utilizing custom algorithms or software that are central to the research but not yet described in published literature, software must be made available to editors and reviewers. We strongly encourage code deposition in a community repository (e.g. GitHub). See the Nature Research guidelines for submitting code & software for further information.

## Data

Policy information about availability of data

All manuscripts must include a data availability statement. This statement should provide the following information, where applicable:
- Accession codes, unique identifiers, or web links for publicly available datasets
- A list of figures that have associated raw data
- A description of any restrictions on data availability

Raw data from Figures 1-4 is available in an Excel file format as Source data. Codon-optimised DNA sequences are deposited in GenBank under the Accession codes OK352257 (cifA) and OK352258 (cifB).

# Field-specific reporting

Please select the one below that is the best fit for your research. If you are not sure, read the appropriate sections before making your selection.

☒ Life sciences    ☐ Behavioural & social sciences    ☐ Ecological, evolutionary & environmental sciences

For a reference copy of the document with all sections, see nature.com/documents/nr-reporting-summary-flat.pdf

# Life sciences study design

All studies must disclose on these points even when the disclosure is negative.

| | |
|---|---|
| Sample size | We conducted power analysis in G*Power 3.1 to detect a 50% reduction in fertility yielding n=5, non-centrality parameter = 21.3, critical chi-squared result of 11.07, total samples size =10 to estimate the sample size required for detecting differences. On average we used far greater sample sizes than this power analysis suggested, as we planned to use more stringent tests for non-parametric data (which cannot be estimated by power analysis). |
| Data exclusions | No data were excluded. |
| Replication | All fertility and egg development phenotypes observed in different groups were confirmed in a minimum of two experiments to ensure reproducibility. Observations were consistent between replicates. Embryo imaging experiments were performed only once as the number of embryos obtained is representative of a large sample size. While fixing and imaging of ovaries was performed only once, observations of ovarian defects were consistent among dissected females in three replicates. |
| Randomization | As organisms were taken from different transgenic colonies to compare during experiments, randomization of individuals was not possible, although care was taken to standardize rearing between colonies. |
| Blinding | Groups were not blinded but were rotated between researchers for different replicates to reduce bias of data collection. |

# Reporting for specific materials, systems and methods

We require information from authors about some types of materials, experimental systems and methods used in many studies. Here, indicate whether each material, system or method listed is relevant to your study. If you are not sure if a list item applies to your research, read the appropriate section before selecting a response.

## Materials & experimental systems

| n/a | Involved in the study |
|---|---|
| ☒ ☐ | Antibodies |
| ☒ ☐ | Eukaryotic cell lines |
| ☒ ☐ | Palaeontology and archaeology |
| ☐ ☒ | Animals and other organisms |
| ☒ ☐ | Human research participants |
| ☒ ☐ | Clinical data |
| ☒ ☐ | Dual use research of concern |

## Methods

| n/a | Involved in the study |
|---|---|
| ☒ ☐ | ChIP-seq |
| ☒ ☐ | Flow cytometry |
| ☒ ☐ | MRI-based neuroimaging |

# Animals and other organisms

Policy information about studies involving animals; ARRIVE guidelines recommended for reporting animal research

| | |
|---|---|
| Laboratory animals | Anopheles gambiae mosquitoes were used from the G3 strain. Species was confirmed using specific primers (Santolamazza et al. 2008, Malar J.). Both males and females were used for experiments. For fertility and egg development experiments, 6-8 day-old females were blood fed and allowed to oviposit. For RNA samples, 3-6 day-old males and females were dissected. |
| Wild animals | There were no wild animals in this study |
| Field-collected samples | There were no field-collected samples in this study |
| Ethics oversight | Approval was received by the Committee on Microbial Safety to create transgenic mosquitoes with Wolbachia-derived genetic sequences |

Note that full information on the approval of the study protocol must also be provided in the manuscript.

