## [Peer Review File. · Nature Microbiology]

Peer Review Information

Journal: Nature Microbiology

Manuscript Title: Wolbachia cifB induces cytoplasmic incompatibility in the malaria mosquito vector

Corresponding author name(s): Flaminia Catteruccia

Reviewer Comments & Decisions:

Decision Letter, initial version:
--

Dear Professor Catteruccia,

Thank you very much for your enquiry about submitting your manuscript "Wolbachia cifB induces cytoplasmic incompatibility in the malaria mosquito" to Nature Microbiology{redacted}

We would be happy to send the full manuscript out for formal review. My advice is to focus on the CI interaction and how your experiments address this important aspect of the insect-Wolbachia interaction. It might also be useful to update the Introduction and discussion with a sense of how these findings fit with the set of published trials from various groups working on Wolbachia, as you hint at screening for mutants or generating mutants that better colonize the insect but I think its important to first make it clear WHY this is a need for control and second discuss what your findings might mean for published trials (is there any knowledge you uncover that explains trials that work well (eg. for Aedes <https://www.nature.com/articles/s41586-019-1407-9>) versus the issues you outline for Anopheles?

In order to submit your complete manuscript to Nature Microbiology, please use the link below:

{redacted}

If you have any questions, please feel free to contact me.

Yours sincerely,

{redacted}

Decision Letter, first revision:

Dear Professor Catteruccia,

Thank you for your patience while your manuscript "Wolbachia cifB induces cytoplasmic incompatibility in the malaria mosquito" was under peer-review at Nature Microbiology. It has now been seen by 3 referees, whose expertise and comments you will find at the end of this email. Although they find your work of some potential interest, they have raised a number of concerns that will need to be addressed before we can consider publication of the work in Nature Microbiology.

In particular, the context setting and attribution of concepts and ideas to other researchers is criticized and must be addressed. Additional experimental evidence is needed to support the conclusion that cifA attenuates cifB mod, and you should test if VASA-cifA is also capable of CI.

Should further experimental data allow you to address these criticisms, we would be happy to look at a revised manuscript.

Please include a data availability statement as a separate section after Methods but before references, under the heading "Data Availability". This section should inform readers about the availability of the data used to support the conclusions of your study. This information includes accession codes to public repositories (data banks for protein, DNA or RNA sequences, microarray, proteomics data etc...), references to source data published alongside the paper, unique identifiers such as URLs to data repository entries, or data set DOIs, and any other statement about data availability. At a minimum, you should include the following statement: "The data that support the findings of this study are available from the corresponding author upon request", mentioning any restrictions on availability. If DOIs are provided, we also strongly encourage including these in the Reference list (authors, title, publisher (repository name), identifier, year). For more guidance on how to write this section please see:

<http://www.nature.com/authors/policies/data/data-availability-statements-data-citations.pdf>

* Include a "Response to referees" document detailing, point-by-point, how you addressed each

referee comment. If no action was taken to address a point, you must provide a compelling argument. This response will be sent back to the referees along with the revised manuscript.

* If you have not done so already we suggest that you begin to revise your manuscript so that it conforms to our Article format instructions at <http://www.nature.com/nmicrobiol/info/final-submission>. Refer also to any guidelines provided in this letter.

When submitting the revised version of your manuscript, please pay close attention to our [href="https://www.nature.com/nature-research/editorial-policies/image-integrity">Digital Image Integrity Guidelines. and to the following points below:](https://www.nature.com/nature-research/editorial-policies/image-integrity)

{redacted}

Note: This url links to your confidential homepage and associated information about manuscripts you may have submitted or be reviewing for us. If you wish to forward this e-mail to co-authors, please delete this link to your homepage first.

Nature Microbiology is committed to improving transparency in authorship. As part of our efforts in this direction, we are now requesting that all authors identified as 'corresponding author' on published papers create and link their Open Researcher and Contributor Identifier (ORCID) with their account on the Manuscript Tracking System (MTS), prior to acceptance. This applies to primary research papers only. ORCID helps the scientific community achieve unambiguous attribution of all scholarly contributions. You can create and link your ORCID from the home page of the MTS by clicking on 'Modify my Springer Nature account'. For more information please visit www.springernature.com/orcid.

If you wish to submit a suitably revised manuscript we would hope to receive it within 6 months. If you cannot send it within this time, please let us know. We will be happy to consider your revision, even if a similar study has been accepted for publication at Nature Microbiology or published elsewhere (up to a maximum of 6 months).

{redacted}

Reviewer Expertise:

Referee #2: Evolutionary genetics

Referee #3: Evolution of host-parasite interactions

Referee #4: Wolbachia-insect interactions

Reviewer Comments:

Reviewer #1 (Remarks to the Author):

This paper represents a relatively small but interesting advance in Wolbachia CI genetics. The authors demonstrate that transgenic expression of the WO prophage-associated gene cifB in Anopheles mosquitoes produces CI. This is in contrast to results from Drosophila that have found both cifB and cifA are required to induce CI. The authors claim their results support the toxin-antidote model of CI proposed by Beckmann and colleagues. They discuss how their results are important for Wolbachia colonization, and specifically in vector systems. I have several major comments about the findings and interpretations that I hope are useful for the authors.

Major comments:

1. The authors claim their results support the toxin-antidote (TA) model of CI, but they do not formally test the specific predictions of this or other models of CI. The main data used as support for the TA model are that a) cifB-only lines could not be generated and b) cifB expression in females reduces fertility. Thus, cifB is toxic. Indeed, that's a good interpretation, but TA and host-modification (HM) models (which the authors do not acknowledge) both agree that the "toxin" or the "mod factor" can be toxic. The models disagree in how that factor contributes to toxicity and how it is prevented. In TA, the toxin acts in the embryo and is prevented by binding to the antidote. In HM the "toxin" modifies some factor in spermatogenesis and that modification gets reversed or otherwise inhibited by the rescue factor in the embryo. The authors do not test these predictions or give time to other models in a way that would make the detailed discussion on this topic useful. It should be removed from the discussion, but the lack of any real test of these models lessens the impact of this work.

2. Discussion of spread/invasion dynamics is deeply confused. CI cannot encourage spread from low frequencies because it is frequency dependent. Wolbachia must increase host fitness in other ways to initially spread from rare frequencies. Once sufficiently common, CI pushes infections to higher frequencies balanced by imperfect transmission. Turelli's work with Hoffmann (Hoffmann et al. 1990, Kriesner et al. 2016), recent papers with Cooper (Meany et al. 2020, Evolution eg), and many others (Turelli 1994, Barton and Turelli, e.g.) are ignored by the authors. This leads to incorrect claims and confusing speculation; for example, the authors speculate in the discussion that the relative expression of cifs may need to be fine-tuned for Wolbachia to successfully invade. This is odd given that CI is not involved in low frequency spread.

3. The authors report that *cifB* alone can cause rescuable transgenic CI in *Anopheles*, which is the novelty of the paper. While it provides no insights into TA vs other models, it does generate some interesting questions/points of discussion that are not addressed. First, why does this gene set in this system allow for a one-by-one model while other gene sets in other systems do not? Is there something biologically different about mosquitoes and fly reproduction, or about the *cif* genes, that may allow for this? Describing some future directions to explore would be much more useful than speculating about models that aren't tested. Second, since *cifB* is the toxin/mod factor, this sheds doubt on versions of the mistiming model that propose that *cifA* is the primary mistiming factor and needs *cifB* to target host processes, or other similar mechanistic proposals where *cifA* is the "toxin". A cautious discussion of this would improve the paper.

4. Reading the paper one would think that the possibility of a one-by-one model of CI has not been considered before. Beckmann has highlighted this possibility in numerous papers (first in Beckmann and Fallon 2013). Shropshire/Bordenstein also recently discussed this in Shropshire et al. 2021 (Genetics) and elsewhere. The authors should take care to acknowledge all of the work/ideas that others have put into this area. Notably, it is not surprising that the two-by-one model does not apply to all systems.

5. More work is needed to support the conclusion that *cifA* attenuates *cifB* mod. A rescue cross needs to be performed on this strain to fully assess this. If the rescue phenotype is different for this line than for rescue of *cifB* alone then perhaps there is some other fitness effect impacting the relative CI phenotype.

6. The authors report that *cifA* rescue is limited by expression and that they are only able to get transgenic rescue with the VASA line. They do not test if VASA-*cifA* is also capable of CI. This experiment would help round out the study to confirm that *cifA* is not also sufficient for mod.

7. In general, the scholarship should be improved. Areas that introduce/discuss spread dynamics (see above/below), CI models (see above), *cif* disruption (see below), and other topics fail to cite the most important work. This leads to incorrect claims (e.g., about spread dynamics) and a feeling while reading the paper that some aspects are more novel than they actually are.

8. A few things could be amended regarding discussion of the application of this work to biocontrol. The authors discuss how *cifB* males could be released in IIT-like applications. However, since *cifB* males are hard to create this seems unrealistic for release. Instead, discussing how *cifA*;B males might be released would be useful. Additionally, to justify that release of transgenes would work for spread applications, it would be helpful to have a *cifA*;B male x *cifA*;B female cross in the same genetic background and promoter scheme to show that not only can *cifA*;B males cause sterility but that the same genetic background can rescue. Walker et al. 2021 (Current Biology) and the Wolbachia malaria blocking work should be cited and discussed in context of this study.

Additional points:

Line 28-33: The authors introduce spread dynamics incorrectly (see above).

Line 69: Briefly describe the expression pattern of *zpg*. Is it only expressed in germline cells? If so, which stages of gametogenesis is expression known to occur?

Line 87: "as the rescue effect has been shown to be dose-dependent". Dosage dependency and localization were both proposed in the cited paper. Both should be noted here. Also, are VASA and zpg expressed in the exact same cells but at different levels, or is it plausible that they also express in different cells? If the later is possible then that hypothesis should be emphasized.

Line 134: "these results reveal rescue effects possibly caused by maternal deposition of CifA". It seems likely that other host factors vary between groups and contribute to these differences.

Line 135 and 173: "cifB expression in females is therefore highly deleterious". This warrants a discussion since these results also conflict with studies in *D. melanogaster* where cifB expression in females has no effect on hatching relative to cifA or cifA;B expression.

Line 147: "our findings are supportive of a parsimonious toxin-antidote model where CifB is the toxin". There is no direct test/evidence for this claim.

Line 163-165: The authors speculate about mutations breaking cifB, but ignore the classic theoretical work indicating that selection does not act to increase or maintain CI (Turelli 1994), and ignore the best empirical example of this demonstrating cifB disruption (and preservation of cifA) in wMau (Meany et al. 2020). The authors should see Martinez et al. 2021.

Line 168: CI does not influence colonization as described.

Reviewer #3 (Remarks to the Author):

A number of strains of the bacterial symbiont *Wolbachia* have evolved a remarkable drive mechanism termed cytoplasmic incompatibility (CI), whereby crosses between infected males and uninfected females result in few or no viable offspring. The genetic and mechanistic basis of CI has long been a holy grail, and the recent discovery that CI is caused by two linked genes (*cifA* and *cifB*) has revolutionized the field, although we are still far from understanding how these genes work to induce CI in males, and to rescue it in females. *cifA* is sufficient to rescue CI in females, but studies on male induction have yielded conflicting and ambiguous results. It is difficult to clearly recapitulate induction, and some researchers have suggested that both *cifA* and *cifB* are required, while others suggest that *cifB* is sufficient. Because *Wolbachia* is as yet uncultivable, it is very difficult to ascribe function to *Wolbachia* genes, and a relatively recent approach is to create transgenic insects that express *Wolbachia* genes, which comes with its own set of challenges.

Wolbachia infections in *Anopheles* mosquitoes are quite rare, and so it has not been possible to explore the potential of CI in *Anopheles* control. So this manuscript makes two important contributions to the field. First, the authors engineer transgenic *Anopheles gambiae* that express *cifA* and *cifB* from bacteria that naturally infect *Culex* mosquitoes, and are able to recapitulate CI. This is exciting as it opens up the possibility of using CI to control *Anopheles*, for example by releasing transgenic incompatibles males in the field. Second, the authors use controlled crosses involving different combinations of transgenic *cif* genes in males and females to help us understand more about *cif* function. They show that males expressing *cifB* are incompatible, but that *cifB* is quite toxic, such that lines expressing only *cifB* are inviable. *cifB* stability and expression are modulated by the presence of

cifA, which fits well with published biochemical in yeast showing that cifA can neutralize cifB protein.

I think that this paper is an important and useful contribution to the field, and will be of interest to researchers studying cytoplasmic incompatibility and other reproductive manipulations, as well as those interested in gene drive and insect disease vector control. The experiments and figures are clear and I found the manuscript easy to follow. The paper would be strengthened by adding some more explanation for the non-expert, for example, by more clearly explaining the differing cif gene naming systems and the biology of some of the Wolbachia strains (e.g. wMel, wPip, wRec). I think it is also useful to remind that there are many successful Wolbachia infections in nature that do not cause CI, so this is not the only factor preventing infection in Anopheles.

Other comments/suggestions:

Line 15: 'still intensely debated' is a bit strong and dramatic. Cif genes were only identified in 2017, so it is not surprising that the mechanism involving these genes is not fully unresolved. One lab strongly promotes the 'two-by-one' language and interpretation, and a few other labs contest this, so there are not enough active players to call it 'intense debate'. [I think of 'two-by-one' as an observation that both cifA and cifB are required for CI in Drosophila, more than a model.]

Line 24-5: I think it is misleading to imply that Wolbachia is rare in Anopheles because of CI. There are likely hundreds of thousands of successful Wolbachia infections in nature involving strains that do not cause/express CI.

Line 28: This is a bit misleading as many Wolbachia strains do not appear to be reproductive manipulators. Wolbachia's success is likely due to many things, including a high affinity for germline.

Line 34: It would be useful to add a few words for non-experts, explaining the different names for CI genes (cif, cid, cin etc...). (Some of these different names are introduced later, but it would be much clearer to briefly explain this up front.)

Line 36: 'disputed' is a bit strong

Line 42: It would be useful to explain to non-experts that wMel is a strain from Drosophila melanogaster flies that does not cause strong CI in its native host.

Line 69: It would be useful to explain to non-experts that wPip is a strain from Culex pipiens mosquitoes that causes strong CI in its native host.

Line 149: It would be useful to explain to non-experts that wRec is a strain from Drosophila recens flies that causes strong CI in its native host.

Reviewer #4 (Remarks to the Author):

Wolbachia is an extraordinarily ecologically symbiont that infects about half of arthropod species, and this success is thought to be in a large part due to its ability to induce cytoplasmic incompatibility. This has been studied for over 50 years, but there has been a resurgence of interest in recent years as

Wolbachia has been deployed to prevent mosquitoes transmitting dengue virus. The recent discovery of the genes underlying this trait was a major milestone on this field, and new insights in this area therefore have the potential to be of broad interest.

The headline result here is *cifB* expression in males is sufficient to induce cytoplasmic incompatibility. This is in line with classical models of the trait, but is different from *Drosophila* where there is strong evidence for *cifA* also being required (this protein 'rescues' the trait in females). This is an important insight. The second main result is that you can induce CI in *Anopheles* mosquitoes, but this is sensitive to the expression level of the factors involved. This will be important both for any application of Wolbachia in *Anopheles* to combat malaria, and also for understanding what might shape Wolbachia transfers between species in nature. A weakness of the paper is that there are no new insights into the molecular mechanisms of these effects, but in my judgement the results stand in the absence of this. The manuscript is clearly written and the results are straightforward. I have some minor suggestions.

The observation that *cifB* is likely toxic but this toxicity is reduced by *cifA* is important, but the results supporting this were sometimes rather anecdotal. Line 72. 'but none that expressed *cifB* only, a suggestion that *cifB* may cause embryonic toxicity alleviated by *cifA* co-expression'. The same result in line 94 is repeated. There is no data shown for this in the results making this a weak way to infer lethality. I suggest this is deleted, or failing that it is presented as a properly analysed experiment. The same conclusion about *cifB* lethality is mentioned later when *cifB* males are generated. Is it not straightforward to present some numbers to support the conclusion here?

Line 62. It is important to cite a critique of the presence of Wolbachia in many species (Chrostek mBio) to acknowledge that some in the field are sceptical of these results. On the other hand, the absence of a citation for the preprint 'Genomic and microscopic evidence of stable high density and maternally inherited Wolbachia infections in *Anopheles* mosquitoes' is odd at this point, as this clearly shows Wolbachia does exist in some species at meaningful levels (I note this is cited later in a different context).

Line 20 "We report that CI can be fully recapitulated in these mosquitoes, and that *cifB* is sufficient to cause this reproductive manipulation". This is confusing as the reproductive manipulation implies CI, which includes rescue, and therefore needs *cifA*. Reword.

Line 101. Delete the subjective word 'comparable' (I look at this graph and see 'less than'). Instead just state the mean hatch rate or something.

Figure 3C and the conclusions following from it. This assumes the two promoters drive expression in precisely the same cell types and stages of germline development. Is there evidence for this?

The Data Availability statement is not acceptable. It does not say where this data will be deposited. Sequence and annotation of constructs should be provided.

Author Rebuttal, first revision:

Reviewer #1 (Remarks to the Author):

This paper represents a relatively small but interesting advance in Wolbachia CI genetics. The authors demonstrate that transgenic expression of the WO prophage-associated gene cifB in Anopheles mosquitoes produces CI. This is in contrast to results from Drosophila that have found both cifB and cifA are required to induce CI. The authors claim their results support the toxin-antidote model of CI proposed by Beckmann and colleagues. They discuss how their results are important for Wolbachia colonization, and specifically in vector systems. I have several major comments about the findings and interpretations that I hope are useful for the authors.

We thank the reviewer for providing useful comments and suggestions that have increased the quality and interest of the revised manuscript.

Major comments:

1. The authors claim their results support the toxin-antidote (TA) model of CI, but they do not formally test the specific predictions of this or other models of CI. The main data used as support for the TA model are that a) cifB-only lines could not be generated and b) cifB expression in females reduces fertility. Thus, cifB is toxic. Indeed, that's a good interpretation, but TA and host-modification (HM) models (which the authors do not acknowledge) both agree that the "toxin" or the "mod factor" can be toxic. The models disagree in how that factor contributes to toxicity and how it is prevented. In TA, the toxin acts in the embryo and is prevented by binding to the antidote. In HM the "toxin" modifies some factor in spermatogenesis and that modification gets reversed or otherwise inhibited by the rescue factor in the embryo. The authors do not test these predictions or give time to other models in a way that would makes the detailed discussion on this topic useful. It should be removed from the discussion, but the lack of any real test of these models lessens the impact of this work.

The reviewer is correct as indeed we did not test whether the infertility induced by *cifB* is caused by this factor acting in the embryo or during spermatogenesis. In this revised version we have modified both our introduction and discussion to take more time to address all proposed models, outlining the early TA predictions and how new evidence contributed to the evolution of these models, leading to the two-by-one and HM models. In our discussion we now clearly state that our findings do not exclude the possibility that modification of sperm occurs prior to sperm transfer (as in the HM model).

2. Discussion of spread/invasion dynamics is deeply confused. CI cannot encourage spread from low frequencies because it is frequency dependent. Wolbachia must increase host fitness in other ways to initially spread from rare frequencies. Once sufficiently common, CI pushes infections to

higher frequencies balanced by imperfect transmission. Turelli's work with Hoffmann (Hoffmann et al. 1990, Kriesner et al. 2016), recent papers with Cooper (Meany et al. 2020, Evolution eg), and many others (Turelli 1994, Barton and Turelli, e.g.) are ignored by the authors. This leads to incorrect claims and confusing speculation; for example, the authors speculate in the discussion that the relative expression of cifs may need to be fine-tuned for Wolbachia to successfully invade. This is odd given that CI is not involved in low frequency spread.

We agree that in places our writing implied that CI is beneficial to *Wolbachia* during initial host colonization while this is not the case. We have now corrected this issue within the manuscript, citing references such as Turelli 1994, Turelli 2010, Hoffmann and Turelli 1991.

3. The authors report that cifB alone can cause rescuable transgenic CI in Anopheles, which is the novelty of the paper. While it provides no insights into TA vs other models, it does generate some interesting questions/points of discussion that are not addressed. First, why does this gene set in this system allow for a one-by-one model while other gene sets in other systems do not? Is there something biologically different about mosquitoes and fly reproduction, or about the cif genes, that may allow for this? Describing some future directions to explore would be much more useful than speculating about models that aren't tested. Second, since cifB is the toxin/mod factor, this sheds doubt on versions of the mistiming model that propose that cifA is the primary mistiming factor and needs cifB to target host processes, or other similar mechanistic proposals where cifA is the "toxin". A cautious discussion of this would improve the paper.

We have added sentences to the discussion describing how expression of wMel cif genes in *An. gambiae* (and further attempts to study wPip Type I cifB expression in *D. melanogaster*) could clarify some of the discrepancies in these systems and offer insight into the possibility of strain- and host-dependent differences in CI induction. We also offer some speculation that other gene sets may in fact also operate on a one-by-one model (see discussion about wRec cifB and wPip Type IV cifB). We also now include a brief discussion of the mistiming model, explaining its inconsistencies with our findings.

4. Reading the paper one would think that the possibility of a one-by-one model of CI has not been considered before. Beckmann has highlighted this possibility in numerous papers (first in Beckmann and Fallon 2013). Shropshire/Bordenstein also recently discussed this in Shropshire et al. 2021 (Genetics) and elsewhere. The authors should take care to acknowledge all of the work/ideas that others have put into this area. Notably, it is not surprising that the two-by-one model does not apply to all systems.

We apologize if in our initial submission we did not stress enough that the one-by-one model has been proposed by others, as this was certainly not our intention. In our amended manuscript we have laid out the evolution of CI models, beginning with the TA model (which, in its original form, could also be described as a one-by-one model) for which we have now cited Beckmann and

Fallon 2013, in addition to our previous citations of the later publications from Beckmann et al. that describe the model in more detail. We later cite Shropshire et al. 2020 (Genetics) and give emphasis to the results of this study that are pertinent to this manuscript in the discussion.

5. More work is needed to support the conclusion that *cifA* attenuates *cifB* mod. A rescue cross needs to be performed on this strain to fully assess this. If the rescue phenotype is different for this line than for rescue of *cifB* alone then perhaps there is some other fitness effect impacting the relative CI phenotype.

We thank the reviewer for this recommendation which has strengthened our work. We have performed these additional experiments and shown that indeed the partial CI induced by these males is rescued by *vasa-cifA* in females (see **Extended Data Figure 3** below). This result is consistent with our presentation and interpretation of the data within the manuscript.

6. The authors report that *cifA* rescue is limited by expression and that they are only able to get transgenic rescue with the *VASA* line. They do not test if *VASA-cifA* is also capable of CI. This experiment would help round out the study to confirm that *cifA* is not also sufficient for mod.

Following this suggestion we have performed crosses to compare fertility of wild-type females crossed with either *vasa-cifA* expressing males or wild-type males. We see that *vasa-cifA* males do not induce more infertility than wild-type males (See **Extended Data Figure 3** below). This result is consistent with our presentation and interpretation of the data within the manuscript.

Extended Data Figure 3: *vasa-cifA* does not cause CI, and likely inhibits its induction when co-expressed in males. *vasa-cifA* expression alone in males does not cause infertility, and its

expression in females is sufficient to rescue the intermediate infertility phenotype caused by expression of *vasa-cifA;zpg-cifB* in males. (Dunn's multiple comparisons, $p \leq 0.0001$ for differences between all statistical groups). Median and interquartile ranges are shown. For each group (top to bottom) the n is as follows: 30, 29, 28, 28. Kruskal-Wallis results: $H=62.87$, $p < 0.0001$, $df=3$.

7. In general, the scholarship should be improved. Areas that introduce/discuss spread dynamics (see above/below), CI models (see above), cif disruption (see below), and other topics fail to cite the most important work. This leads to incorrect claims (e.g., about spread dynamics) and a feeling while reading the paper that some aspects are more novel than they actually are.

We thank the reviewer for pointing out areas where we can strengthen our discussion. As addressed above, we agree that our writing implied some incorrect notions regarding *Wolbachia*'s spread dynamics and have made efforts to clarify these concepts within the manuscript. Similarly, as addressed above, we have elaborated upon our discussion of different models for CI. Regarding *cif* gene disruption, as we elaborate on below, we hope that this is clarified also by an inclusion of the statement that *cifB* pseudogenization is not uncommon within insects (Martinez et al., 2021), as expected by the evolutionary theory described by Turelli et al. 1994.

8. A few things could be amended regarding discussion of the application of this work to biocontrol. The authors discuss how cifB males could be released in IIT-like applications. However, since cifB males are hard to create this seems unrealistic for release. Instead, discussing how cifA;B males might be released would be useful. Additionally, to justify that release of transgenes would work for spread applications, it would be helpful to have a cifA;B male x cifA;B female cross in the same genetic background and promoter scheme to show that not only can cifA;B males cause sterility but that the same genetic background can rescue. Walker et al. 2021 (Current Biology) and the Wolbachia malaria blocking work should be cited and discussed in context of this study.

The reviewer has good points, and this discussion was underdeveloped around applications of *cif* genes for IIT-like releases etc. We have added *cifA;B* males as other candidates for IIT. In reality, it is difficult to rear even *cifA;B* males, and likely if this strategy was used, conditional expression such as under a Tet On/Off system might be needed, as we have now noted this in the discussion. Certainly, many more studies would be required to determine the safety and achievability of such a strategy.

We did not intend to suggest that *cifA* and *cifB* can be directly used for spread applications (similar to population replacement gene drives), as these would rely on good infertility and rescue induced by the same transgene couple in males and females, respectively, which we did not observe in this study. We believe our current promoter expression systems would need to be optimized for spread, and further exploration of different promoters would be necessary to drive such a system in real-world applications. As such, we have left this possible application out of our discussion,

but we agree that our data provides an interesting proof of principle upon which such future studies can build.

We have now added a brief sentence referring to the pathogen protection effects observed by *Wolbachia* on *Plasmodium*, and have mentioned the study by Walker et al. 2021 in both the introduction and the discussion, where we have noted its discovery of *cif* genes in *Wolbachia* infections of *Anopheles*.

Additional points:

Line 28-33: The authors introduce spread dynamics incorrectly (see above).

We have changed the wording to note that we refer to CI-inducing strains, and colonization of *Anopheles*. We do not elaborate on our logic here in the abstract in the interest of succinctness, but later explain that if CifB imposes costly effects on reproduction (including male incompatibility) that are not fully rescued by CifA, it may be difficult to overcome these effects, and thus CI (or *Wolbachia* itself) might be lost during host colonization. We do not intend to argue that the lack of CI per se would lead to the failure of *Wolbachia* to colonize and potentially even spread, however we argue that dysregulated expression of *cifB*, if not fully rescued, would be detrimental to *Wolbachia*'s persistence. Thus, *Wolbachia* must finely tune its expression of this toxic gene (and its rescue factor) for CI to be maintained.

*Line 69: Briefly describe the expression pattern of *zpg*. Is it only expressed in germline cells? If so, which stages of gametogenesis is expression known to occur?*

While both *zpg* and *vasa* expression is largely limited to the germline, in flies, *vasa* expression is found in somatic gonadal precursor cells in both females and males (Renault et al. 2012). It is unclear if this occurs in *Anopheles*. We now include these details in the manuscript.

*Line 87: "as the rescue effect has been shown to be dose-dependent". Dosage dependency and localization were both proposed in the cited paper. Both should be noted here. Also, are VASA and *zpg* expressed in the exact same cells but at different levels, or is it plausible that they also express in different cells? If the later is possible then that hypothesis should be emphasized.*

We have now changed this sentence to read "the rescue effect has been shown to be promoter-dependent". As mentioned above, in flies there are subtle differences in expression patterns of *vasa* and *zpg*, and we cannot exclude that these, rather than expression level, could contribute to differences in CI and rescue. We have added a statement and changed some of the wording in the manuscript to reflect this.

Line 134: “these results reveal rescue effects possibly caused by maternal deposition of CifA”. It seems likely that other host factors vary between groups and contribute to these differences.

In mosquitoes expressing GFP under the *vasa* promoter, maternal deposition of GFP was observed in all of their offspring (not just those that inherited the transgene), so what we propose is not farfetched (Papathanos et al. 2009). There is evidence that this occurs either to a lesser extent or not at all for transcripts under the *zpg* promoter (Hammond et al. 2020, BioRxiv). We have now briefly referred to this evidence when describing the results.

Based on our analysis of female offspring expressing *cifB*, large differences in both fecundity and fertility were observed between *cifB* females derived from *zpg-cifA*-expressing mothers compared to *vasa-cifA*-expressing mothers. Furthermore, when no maternal *cifA* was contributed (in the *v^{pat}zpg-cifB* females), no rescue was seen. This is particularly striking given that, in this experiment, the genetic contributions are otherwise the same (one WT parent and one *vasa-cifA;zpg-cifB* parent). All females used in the experiment are thus the result of outcrossing transgenic lines with WT individuals, therefore inbreeding of transgenic lines is also unlikely to be playing a role here. We thus conclude maternal effects are most likely to cause these differences. Following this reviewer’s comment, we nevertheless added a sentence stating that we cannot rule out our host factors may also play a role.

Line 135 and 173: “cifB expression in females is therefore highly deleterious”. This warrants a discussion since these results also conflict with studies in D. melanogaster where cifB expression in females has no effect on hatching relative to cifA or cifA;B expression.

This is indeed another discrepancy between these investigations, and we have added a comment about these different result to the discussion section.

Line 147: “our findings are supportive of a parsimonious toxin-antidote model where CifB is the toxin”. There is no direct test/evidence for this claim.

We have elaborated on these models, and the caveats that remain regarding the TA model. We now state that we do not exclude the mechanism described by the HM model.

Line 163-165: The authors speculate about mutations breaking cifB, but ignore the classic theoretical work indicating that selection does not act to increase or maintain CI (Turelli 1994), and ignore the best empirical example of this demonstrating cifB disruption (and preservation of cifA) in wMau (Meany et al. 2020). The authors should see Martinez et al. 2021.

We thank the reviewer for these recommended publications, and indeed believe that referring to the data within these studies has strengthened our manuscript. In our view, these works are not inconsistent with, but in fact support our arguments here. As they state, mutations in *cifB* may

arise spontaneously throughout evolution. We suggest only that their loss may be accelerated if selection against *cifB* (and thus against CI) occurs due to its potential toxicity to female reproduction. This is certainly speculative, we do not have conclusive evidence supporting the idea that *Anopheles* are hypersensitive to *cifB* compared to other insects and do not know if, in a natural *Wolbachia* infection, *cifA* expression would be sufficient to overcome these effects (as may be the case in most successful CI-inducing *Wolbachia* infections). We now specify that *cifB* pseudogenization occurs commonly (Martinez et al, 2021), and is expected based on evolutionary modeling of CI (Turelli 1994).

Line 168: CI does not influence colonization as described.

We have amended this discussion in line with ideas presented above, and focus on whether *cifB* toxicity could limit the capacity of CI-inducing *Wolbachia* strains to persist in *Anopheles* due to their potential for causing reproductive toxicity.

Reviewer #3 (Remarks to the Author):

A number of strains of the bacterial symbiont Wolbachia have evolved a remarkable drive mechanism termed cytoplasmic incompatibility (CI), whereby crosses between infected males and uninfected females result in few or no viable offspring. The genetic and mechanistic basis of CI has long been a holy grail, and the recent discovery that CI is caused by two linked genes (cifA and cifB) has revolutionized the field, although we are still far from understanding how these genes work to induce CI in males, and to rescue it in females. cifA is sufficient to rescue CI in females, but studies on male induction have yielded conflicting and ambiguous results. It is difficult to clearly recapitulate induction, and some researchers have suggested that both cifA and cifB are required, while others suggest that cifB is sufficient. Because Wolbachia is as yet uncultivable, it is very difficult to ascribe function to Wolbachia genes, and a relatively recent approach is to create transgenic insects that express Wolbachia genes, which comes with its own set of challenges.

Wolbachia infections in Anopheles mosquitoes are quite rare, and so it has not been possible to explore the potential of CI in Anopheles control. So this manuscript makes two important contributions to the field. First, the authors engineer transgenic Anopheles gambiae that express cifA and cifB from bacteria that naturally infect Culex mosquitoes, and are able to recapitulate CI. This is exciting as it opens up the possibility of using CI to control Anopheles, for example by releasing transgenic incompatibles males in the field. Second, the authors use controlled crosses involving different combinations of transgenic cif genes in males and females to help us understand more about cif function. They show that males expressing cifB are incompatible, but that cifB is quite toxic, such that lines expressing only cifB are inviable. cifB stability and

expression are modulated by the presence of *cifA*, which fits well with published biochemical in yeast showing that *cifA* can neutralize *cifB* protein.

*I think that this paper is an important and useful contribution to the field, and will be of interest to researchers studying cytoplasmic incompatibility and other reproductive manipulations, as well as those interested in gene drive and insect disease vector control. The experiments and figures are clear and I found the manuscript easy to follow. The paper would be strengthened by adding some more explanation for the non-expert, for example, by more clearly explaining the differing *cif* gene naming systems and the biology of some of the *Wolbachia* strains (e.g. *wMel*, *wPip*, *wRec*). I think it is also useful to remind that there are many successful *Wolbachia* infections in nature that do not cause CI, so this is not the only factor preventing infection in *Anopheles*.*

We thank the reviewer for finding this paper an important contribution to the field, and for their thoughtful comments. We have followed their suggestions and have provided more explanation for non-experts, including a brief explanation of the nomenclature for CI genes. We have also clarified that there are *Wolbachia* strains that do not cause CI but are still successful at invading insect populations, as also suggested by another reviewer.

Other comments/suggestions:

*Line 15: 'still intensely debated' is a bit strong and dramatic. Cif genes were only identified in 2017, so it is not surprising that the mechanism involving these genes is not fully unresolved. One lab strongly promotes the 'two-by-one' language and interpretation, and a few other labs contest this, so there are not enough active players to call it 'intense debate'. [I think of 'two-by-one' as an observation that both *cifA* and *cifB* are required for CI in *Drosophila*, more than a model.]*

We have changed “intensely debated” to “unclear”, as it is true it is still a relatively new discovery and thus unsurprising that it constitutes an unresolved question.

*Line 24-5: I think it is misleading to imply that *Wolbachia* is rare in *Anopheles* because of CI. There are likely hundreds of thousands of successful *Wolbachia* infections in nature involving strains that do not cause/express CI.*

We agree with the reviewer, and as mentioned above we have softened this idea by saying “contributed to” rather than “explaining”, and specified that this would only apply to CI-inducing *Wolbachia*.

*Line 28: This is a bit misleading as many *Wolbachia* strains do not appear to be reproductive manipulators. *Wolbachia*'s success is likely due to many things, including a high affinity for germline.*

We thank the reviewer for pointing this out and have now reworded this section to specify that *Wolbachia* are successful insect colonizers, and some strains are also remarkable for their ability to manipulate reproduction, without necessarily linking the two.

Line 34: It would be useful to add a few words for non-experts, explaining the different names for CI genes (cif, cid, cin etc...). (Some of these different names are introduced later, but it would be much clearer to briefly explain this up front.)

We have added brief descriptions of the different nomenclature for *cif* vs *cid*.

Line 36: 'disputed' is a bit strong

We have changed this word to “debated”.

Line 42: It would be useful to explain to non-experts that wMel is a strain from Drosophila melanogaster flies that does not cause strong CI in its native host.

We agree this detail is useful, but believe it is more relevant to mention later in the discussion, so we include it there instead.

Line 69: It would be useful to explain to non-experts that wPip is a strain from Culex pipiens mosquitoes that causes strong CI in its native host.

We have added these details.

Line 149: It would be useful to explain to non-experts that wRec is a strain from Drosophila recens flies that causes strong CI in its native host.

We have added a short note of this nature.

Reviewer #4 (Remarks to the Author):

Wolbachia is an extraordinarily ecologically symbiont that infects about half of arthropod species, and this success is thought to be in a large part due to its ability to induce cytoplasmic incompatibility. This has been studied for over 50 years, but there has been a resurgence of interest in recent years as Wolbachia has been deployed to prevent mosquitoes transmitting dengue virus. The recent discovery of the genes underlying this trait was a major milestone on this field, and new insights in this area therefore have the potential to be of broad interest.

The headline result here is cifB expression in males is sufficient to induce cytoplasmic incompatibility. This is in line with classical models of the trait, but is different from Drosophila where there is strong evidence for cifA also being required (this protein ‘rescues’ the trait in females). This is an important insight. The second main result is that you can induce CI in Anopheles mosquitoes, but this is sensitive to the expression level of the factors involved. This will be important both for any application of Wolbachia in Anopheles to combat malaria, and also for understanding what might shape Wolbachia transfers between species in nature. A weakness of the paper is that there are no new insights into the molecular mechanisms of these effects, but in my judgement the results stand in the absence of this. The manuscript is clearly written and the results are straightforward. I have some minor suggestions.

We are grateful to the reviewer for their positive considerations of our work.

The observation that cifB is likely toxic but this toxicity is reduced by cifA is important, but the results supporting this were sometimes rather anecdotal. Line 72. ‘but none that expressed cifB only, a suggestion that cifB may cause embryonic toxicity alleviated by cifA co-expression’. The same result in line 94 is repeated. There is no data shown for this in the results making this a weak way to infer lethality. I suggest this is deleted, or failing that it is presented as a properly analysed experiment. The same conclusion about cifB lethality is mentioned later when cifB males are generated. Is it not straightforward to present some numbers to support the conclusion here?

We agree with the reviewer that in the original format the data presented was quite anecdotal and have removed these comments from the results. Additionally, we have added some data showing the number of embryos injected in each experiment in the methods section.

Line 62. It is important to cite a critique of the presence of Wolbachia in many species (Chrostek mBio) to acknowledge that some in the field are sceptical of these results. On the other hand, the absence of a citation for the preprint ‘Genomic and microscopic evidence of stable high density and maternally inherited Wolbachia infections in Anopheles mosquitoes’ is odd at this point, as this clearly shows Wolbachia does exist in some species at meaningful levels (I note this is cited later in a different context).

We now mention the detection of *Wolbachia* in *Anopheles* has been criticized for its robustness (Chrostek et al. 2019), although we personally believe that the higher titer *Wolbachia* infections in *An. moucheti* and *An. demeilloni* (now published) by Walker et al. 2021 support previous findings showing that *Wolbachia* can infect anophelines. We now cite Walker et al. in the introduction in addition to the discussion.

Line 20 “We report that CI can be fully recapitulated in these mosquitoes, and that cifB is sufficient

to cause this reproductive manipulation". This is confusing as the reproductive manipulation implies CI, which includes rescue, and therefore needs cifA. Rerword.

Throughout the manuscript we have addressed this when relevant, specifying that we intend to describe the embryonic lethality associated with CI.

Line 101. Delete the subjective word 'comparable' (I look at this graph and see 'less than'). Instead just state the mean hatch rate or something.

We have removed the word 'comparable' and described the data more objectively.

Figure 3C and the conclusions following from it. This assumes the two promoters drive expression in precisely the same cell types and stages of germline development. Is there evidence for this?

Subtle differences between expression patterns of *vasa* and *zpg* exist in flies, and we have now included this information when we introduce the *vasa* promoter. We understand the reviewer's concerns that differences beyond expression level could contribute to the different induction of infertility between the *zpg-cifA; zpg-cifB* and *vasa-cifA; zpg-cifB* males. We have now elaborated on other possibilities for this different rescue within the manuscript, and changed the beginning of the results section to reflect the possibility that differences in timing or localization of *zpg* and *vasa* promoter activity may be contributing to the different infertility phenotypes. Due to the different presentation of these results Fig. 3c and Fig. 3d have been swapped.

The Data Availability statement is not acceptable. It does not say where this data will be deposited. Sequence and annotation of constructs should be provided.

We apology for this erroneous omission, which we now have corrected. The sequencing and annotation of constructs will be certainly made available and accessible, and we have specified that this information will be available on GenBank

Decision Letter, second revision:

Dear Professor Catteruccia

Thank you for submitting the revised version of your Article entitled "Wolbachia cifB induces cytoplasmic incompatibility in the malaria mosquito" for consideration. Thanks also for your patience whilst I read the revision, considered the points made in the response to reviewers and sought input from the team.

I regret to inform you that after careful consideration and discussion with my editorial colleagues, we have decided that we will not be sending the manuscript back to our referees and will no longer be able to consider it for publication in Nature Microbiology.

Although one aim of the submitted work is to evaluate whether cifB can induce sterility in malaria mosquitoes, the paper as written is more closely focused on understanding which model for cifA/B or cidA/B function is correct. The issues over which model is correct seem to us to be of specialist interest (it is feasible both models may be correct and that cifA/B might function differently depending on the host insect). Work towards malaria vector control is within scope, and of interest to the team, however, in the context of the complete and revised manuscript, this part of the work seems to us to be too preliminary with regards to validating this approach for us to proceed.

We appreciate the revisions made and the additional work included in the latest version of the manuscript, and are not questioning whether the work is sufficient, rather we are not convinced that the malaria control aspect is either central enough to the paper or well-developed enough, to further consider your manuscript for our journal.

We hope that you will rapidly receive a more favourable response elsewhere and would be happy to engage with our colleagues at Nature Communications to assess whether they can proceed and send your manuscript back to reviewers.

I am sorry that we cannot respond more positively on this occasion.

Yours sincerely

{ redacted }

Author Rebuttal, Second Revision

Reviewer #1 (Remarks to the Author):

This paper represents a relatively small but interesting advance in Wolbachia CI genetics. The authors demonstrate that transgenic expression of the WO prophage-associated gene cifB in Anopheles mosquitoes produces CI. This is in contrast to results from Drosophila that have found both cifB and cifA are required to induce CI. The authors claim their results support the toxin-antidote model of CI proposed by Beckmann and colleagues. They discuss how their results are important for Wolbachia colonization, and specifically in vector systems. I have several major comments about the findings and interpretations that I hope are useful for the authors.

We thank the reviewer for providing useful comments and suggestions that have increased the quality and interest of the revised manuscript.

Major comments:

1. *The authors claim their results support the toxin-antidote (TA) model of CI, but they do not*

formally test the specific predictions of this or other models of CI. The main data used as support for the TA model are that a) cifB-only lines could not be generated and b) cifB expression in females reduces fertility. Thus, cifB is toxic. Indeed, that's a good interpretation, but TA and host-modification (HM) models (which the authors do not acknowledge) both agree that the "toxin" or the "mod factor" can be toxic. The models disagree in how that factor contributes to toxicity and how it is prevented. In TA, the toxin acts in the embryo and is prevented by binding to the antidote. In HM the "toxin" modifies some factor in spermatogenesis and that modification gets reversed or otherwise inhibited by the rescue factor in the embryo. The authors do not test these predictions or give time to other models in a way that would makes the detailed discussion on this topic useful. It should be removed from the discussion, but the lack of any real test of these models lessens the impact of this work.

The reviewer is correct as indeed we did not test whether the infertility induced by *cifB* is caused by this factor acting in the embryo or during spermatogenesis. In this revised version we have modified both our introduction and discussion to take more time to address all proposed models, outlining the early TA predictions and how new evidence contributed to the evolution of these models, leading to the two-by-one and HM models. In our discussion we now clearly state that our findings do not exclude the possibility that modification of sperm occurs prior to sperm transfer (as in the HM model).

2. Discussion of spread/invasion dynamics is deeply confused. CI cannot encourage spread from low frequencies because it is frequency dependent. Wolbachia must increase host fitness in other ways to initially spread from rare frequencies. Once sufficiently common, CI pushes infections to higher frequencies balanced by imperfect transmission. Turelli's work with Hoffmann (Hoffmann et al. 1990, Kriesner et al. 2016), recent papers with Cooper (Meany et al. 2020, Evolution eg), and many others (Turelli 1994, Barton and Turelli, e.g.) are ignored by the authors. This leads to incorrect claims and confusing speculation; for example, the authors speculate in the discussion that the relative expression of cifs may need to be fine-tuned for Wolbachia to successfully invade. This is odd given that CI is not involved in low frequency spread.

We agree that in places our writing implied that CI is beneficial to *Wolbachia* during initial host colonization while this is not the case. We have now corrected this issue within the manuscript, citing references such as Turelli 1994, Turelli 2010, Hoffmann and Turelli 1991.

3. The authors report that cifB alone can cause rescuable transgenic CI in Anopheles, which is the novelty of the paper. While it provides no insights into TA vs other models, it does generate some interesting questions/points of discussion that are not addressed. First, why does this gene set in this system allow for a one-by-one model while other gene sets in other systems do not? Is there something biologically different about mosquitoes and fly reproduction, or about the cif genes, that may allow for this? Describing some future directions to explore would be much more useful than speculating about models that aren't tested. Second, since cifB is the toxin/mod factor,

this sheds doubt on versions of the mistiming model that propose that cifA is the primary mistiming factor and needs cifB to target host processes, or other similar mechanistic proposals where cifA is the "toxin". A cautious discussion of this would improve the paper.

We have added sentences to the discussion describing how expression of wMel cif genes in *An. gambiae* (and further attempts to study wPip Type I cifB expression in *D. melanogaster*) could clarify some of the discrepancies in these systems and offer insight into the possibility of strain- and host-dependent differences in CI induction. We also offer some speculation that other gene sets may in fact also operate on a one-by-one model (see discussion about wRec cifB and wPip Type IV cifB). We also now include a brief discussion of the mistiming model, explaining its inconsistencies with our findings.

4. Reading the paper one would think that the possibility of a one-by-one model of CI has not been considered before. Beckmann has highlighted this possibility in numerous papers (first in Beckmann and Fallon 2013). Shropshire/Bordenstein also recently discussed this in Shropshire et al. 2021 (Genetics) and elsewhere. The authors should take care to acknowledge all of the work/ideas that others have put into this area. Notably, it is not surprising that the two-by-one model does not apply to all systems.

We apologize if in our initial submission we did not stress enough that the one-by-one model has been proposed by others, as this was certainly not our intention. In our amended manuscript we have laid out the evolution of CI models, beginning with the TA model (which, in its original form, could also be described as a one-by-one model) for which we have now cited Beckmann and Fallon 2013, in addition to our previous citations of the later publications from Beckmann et al. that describe the model in more detail. We later cite Shropshire et al. 2020 (Genetics) and give emphasis to the results of this study that are pertinent to this manuscript in the discussion.

5. More work is needed to support the conclusion that cifA attenuates cifB mod. A rescue cross needs to be performed on this strain to fully assess this. If the rescue phenotype is different for this line than for rescue of cifB alone then perhaps there is some other fitness effect impacting the relative CI phenotype.

We thank the reviewer for this recommendation which has strengthened our work. We have performed these additional experiments and shown that indeed the partial CI induced by these males is rescued by *vasa-cifA* in females (see **Extended Data Figure 3** below). This result is consistent with our presentation and interpretation of the data within the manuscript.

6. The authors report that cifA rescue is limited by expression and that they are only able to get transgenic rescue with the VASA line. They do not test if VASA-cifA is also capable of CI. This experiment would help round out the study to confirm that cifA is not also sufficient for mod.

Following this suggestion we have performed crosses to compare fertility of wild-type females crossed with either *vasa-cifA* expressing males or wild-type males. We see that *vasa-cifA* males do not induce more infertility than wild-type males (See **Extended Data Figure 3** below). This result is consistent with our presentation and interpretation of the data within the manuscript.

Extended Data Figure 3: *vasa-cifA* does not cause CI, and likely inhibits its induction when co-expressed in males. *vasa-cifA* expression alone in males does not cause infertility, and its expression in females is sufficient to rescue the intermediate infertility phenotype caused by expression of *vasa-cifA; zpg-cifB* in males. (Dunn's multiple comparisons, $p \leq 0.0001$ for differences between all statistical groups). Median and interquartile ranges are shown. For each group (top to bottom) the n is as follows: 30, 29, 28, 28. Kruskal-Wallis results: $H=62.87$, $p < 0.0001$, $df=3$.

7. In general, the scholarship should be improved. Areas that introduce/discuss spread dynamics (see above/below), CI models (see above), *cif* disruption (see below), and other topics fail to cite the most important work. This leads to incorrect claims (e.g., about spread dynamics) and a feeling while reading the paper that some aspects are more novel than they actually are.

We thank the reviewer for pointing out areas where we can strengthen our discussion. As addressed above, we agree that our writing implied some incorrect notions regarding *Wolbachia*'s spread dynamics and have made efforts to clarify these concepts within the manuscript. Similarly, as addressed above, we have elaborated upon our discussion of different models for CI. Regarding *cif* gene disruption, as we elaborate on below, we hope that this is clarified also by an inclusion of the statement that *cifB* pseudogenization is not uncommon within insects (Martinez et al., 2021), as expected by the evolutionary theory described by Turelli et al. 1994.

8. A few things could be amended regarding discussion of the application of this work to biocontrol. The authors discuss how *cifB* males could be released in IIT-like applications. However, since *cifB* males are hard to create this seems unrealistic for release. Instead, discussing how *cifA;B* males might be released would be useful. Additionally, to justify that release of transgenes would work for spread applications, it would be helpful to have a *cifA;B* male x *cifA;B* female cross in the same genetic background and promoter scheme to show that not only can *cifA;B* males cause sterility but that the same genetic background can rescue. Walker et al. 2021 (*Current Biology*) and the *Wolbachia malaria* blocking work should be cited and discussed in context of this study.

The reviewer has good points, and this discussion was underdeveloped around applications of *cif* genes for IIT-like releases etc. We have added *cifA;B* males as other candidates for IIT. In reality, it is difficult to rear even *cifA;B* males, and likely if this strategy was used, conditional expression such as under a Tet On/Off system might be needed, as we have now noted this in the discussion. Certainly, many more studies would be required to determine the safety and achievability of such a strategy.

We did not intend to suggest that *cifA* and *cifB* can be directly used for spread applications (similar to population replacement gene drives), as these would rely on good infertility and rescue induced by the same transgene couple in males and females, respectively, which we did not observe in this study. We believe our current promoter expression systems would need to be optimized for spread, and further exploration of different promoters would be necessary to drive such a system in real-world applications. As such, we have left this possible application out of our discussion, but we agree that our data provides an interesting proof of principle upon which such future studies can build.

We have now added a brief sentence referring to the pathogen protection effects observed by *Wolbachia* on *Plasmodium*, and have mentioned the study by Walker et al. 2021 in both the introduction and the discussion, where we have noted its discovery of *cif* genes in *Wolbachia* infections of *Anopheles*.

Additional points:

Line 28-33: The authors introduce spread dynamics incorrectly (see above).

We have changed the wording to note that we refer to CI-inducing strains, and colonization of *Anopheles*. We do not elaborate on our logic here in the abstract in the interest of succinctness, but later explain that if CifB imposes costly effects on reproduction (including male incompatibility) that are not fully rescued by CifA, it may be difficult to overcome these effects, and thus CI (or *Wolbachia* itself) might be lost during host colonization. We do not intend to argue that the lack of CI per se would lead to the failure of *Wolbachia* to colonize and potentially even spread, however

we argue that dysregulated expression of *cifB*, if not fully rescued, would be detrimental to *Wolbachia*'s persistence. Thus, *Wolbachia* must finely tune its expression of this toxic gene (and its rescue factor) for CI to be maintained.

Line 69: Briefly describe the expression pattern of zpg. Is it only expressed in germline cells? If so, which stages of gametogenesis is expression known to occur?

While both *zpg* and *vasa* expression is largely limited to the germline, in flies, *vasa* expression is found in somatic gonadal precursor cells in both females and males (Renault et al. 2012). It is unclear if this occurs in *Anopheles*. We now include these details in the manuscript.

Line 87: "as the rescue effect has been shown to be dose-dependent". Dosage dependency and localization were both proposed in the cited paper. Both should be noted here. Also, are VASA and zpg expressed in the exact same cells but at different levels, or is it plausible that they also express in different cells? If the later is possible then that hypothesis should be emphasized.

We have now changed this sentence to read "the rescue effect has been shown to be promoter-dependent". As mentioned above, in flies there are subtle differences in expression patterns of *vasa* and *zpg*, and we cannot exclude that these, rather than expression level, could contribute to differences in CI and rescue. We have added a statement and changed some of the wording in the manuscript to reflect this.

Line 134: "these results reveal rescue effects possibly caused by maternal deposition of CifA". It seems likely that other host factors vary between groups and contribute to these differences.

In mosquitoes expressing GFP under the *vasa* promoter, maternal deposition of GFP was observed in all of their offspring (not just those that inherited the transgene), so what we propose is not farfetched (Papathanos et al. 2009). There is evidence that this occurs either to a lesser extent or not at all for transcripts under the *zpg* promoter (Hammond et al. 2020, BioRxiv). We have now briefly referred to this evidence when describing the results.

Based on our analysis of female offspring expressing *cifB*, large differences in both fecundity and fertility were observed between *cifB* females derived from *zpg-cifA*-expressing mothers compared to *vasa-cifA*-expressing mothers. Furthermore, when no maternal *cifA* was contributed (in the *v^{pat}zpg-cifB* females), no rescue was seen. This is particularly striking given that, in this experiment, the genetic contributions are otherwise the same (one WT parent and one *vasa-cifA;zpg-cifB* parent). All females used in the experiment are thus the result of outcrossing transgenic lines with WT individuals, therefore inbreeding of transgenic lines is also unlikely to be playing a role here. We thus conclude maternal effects are most likely to cause these differences. Following this reviewer's comment, we nevertheless added a sentence stating that we cannot rule out our host factors may also play a role.

Line 135 and 173: “cifB expression in females is therefore highly deleterious”. This warrants a discussion since these results also conflict with studies in D. melanogaster where cifB expression in females has no effect on hatching relative to cifA or cifA;B expression.

This is indeed another discrepancy between these investigations, and we have added a comment about these different result to the discussion section.

Line 147: “our findings are supportive of a parsimonious toxin-antidote model where CifB is the toxin”. There is no direct test/evidence for this claim.

We have elaborated on these models, and the caveats that remain regarding the TA model. We now state that we do not exclude the mechanism described by the HM model.

Line 163-165: The authors speculate about mutations breaking cifB, but ignore the classic theoretical work indicating that selection does not act to increase or maintain CI (Turelli 1994), and ignore the best empirical example of this demonstrating cifB disruption (and preservation of cifA) in wMau (Meany et al. 2020). The authors should see Martinez et al. 2021.

We thank the reviewer for these recommended publications, and indeed believe that referring to the data within these studies has strengthened our manuscript. In our view, these works are not inconsistent with, but in fact support our arguments here. As they state, mutations in *cifB* may arise spontaneously throughout evolution. We suggest only that their loss may be accelerated if selection against *cifB* (and thus against CI) occurs due to its potential toxicity to female reproduction. This is certainly speculative, we do not have conclusive evidence supporting the idea that *Anopheles* are hypersensitive to *cifB* compared to other insects and do not know if, in a natural *Wolbachia* infection, *cifA* expression would be sufficient to overcome these effects (as may be the case in most successful CI-inducing *Wolbachia* infections). We now specify that *cifB* pseudogenization occurs commonly (Martinez et al, 2021), and is expected based on evolutionary modeling of CI (Turelli 1994).

Line 168: CI does not influence colonization as described.

We have amended this discussion in line with ideas presented above, and focus on whether *cifB* toxicity could limit the capacity of CI-inducing *Wolbachia* strains to persist in *Anopheles* due to their potential for causing reproductive toxicity.

Reviewer #3 (Remarks to the Author):

A number of strains of the bacterial symbiont Wolbachia have evolved a remarkable drive

mechanism termed cytoplasmic incompatibility (CI), whereby crosses between infected males and uninfected females result in few or no viable offspring. The genetic and mechanistic basis of CI has long been a holy grail, and the recent discovery that CI is caused by two linked genes (cifA and cifB) has revolutionized the field, although we are still far from understanding how these genes work to induce CI in males, and to rescue it in females. cifA is sufficient to rescue CI in females, but studies on male induction have yielded conflicting and ambiguous results. It is difficult to clearly recapitulate induction, and some researchers have suggested that both cifA and cifB are required, while others suggest that cifB is sufficient. Because Wolbachia is as yet uncultivable, it is very difficult to ascribe function to Wolbachia genes, and a relatively recent approach is to create transgenic insects that express Wolbachia genes, which comes with its own set of challenges.

Wolbachia infections in Anopheles mosquitoes are quite rare, and so it has not been possible to explore the potential of CI in Anopheles control. So this manuscript makes two important contributions to the field. First, the authors engineer transgenic Anopheles gambiae that express cifA and cifB from bacteria that naturally infect Culex mosquitoes, and are able to recapitulate CI. This is exciting as it opens up the possibility of using CI to control Anopheles, for example by releasing transgenic incompatibles males in the field. Second, the authors use controlled crosses involving different combinations of transgenic cif genes in males and females to help us understand more about cif function. They show that males expressing cifB are incompatible, but that cifB is quite toxic, such that lines expressing only cifB are inviable. cifB stability and expression are modulated by the presence of cifA, which fits well with published biochemical in yeast showing that cifA can neutralize cifB protein.

I think that this paper is an important and useful contribution to the field, and will be of interest to researchers studying cytoplasmic incompatibility and other reproductive manipulations, as well as those interested in gene drive and insect disease vector control. The experiments and figures are clear and I found the manuscript easy to follow. The paper would be strengthened by adding some more explanation for the non-expert, for example, by more clearly explaining the differing cif gene naming systems and the biology of some of the Wolbachia strains (e.g. wMel, wPip, wRec). I think it is also useful to remind that there are many successful Wolbachia infections in nature that do not cause CI, so this is not the only factor preventing infection in Anopheles.

We thank the reviewer for finding this paper an important contribution to the field, and for their thoughtful comments. We have followed their suggestions and have provided more explanation for non-experts, including a brief explanation of the nomenclature for CI genes. We have also clarified that there are *Wolbachia* strains that do not cause CI but are still successful at invading insect populations, as also suggested by another reviewer.

Other comments/suggestions:

Line 15: 'still intensely debated' is a bit strong and dramatic. Cif genes were only identified in 2017, so it is not surprising that the mechanism involving these genes is not fully unresolved. One lab strongly promotes the 'two-by-one' language and interpretation, and a few other labs contest this, so there are not enough active players to call it 'intense debate'. [I think of 'two-by-one' as an observation that both cifA and cifB are required for CI in Drosophila, more than a model.]

We have changed “intensely debated” to “unclear”, as it is true it is still a relatively new discovery and thus unsurprising that it constitutes an unresolved question.

Line 24-5: I think it is misleading to imply that Wolbachia is rare in Anopheles because of CI. There are likely hundreds of thousands of successful Wolbachia infections in nature involving strains that do not cause/express CI.

We agree with the reviewer, and as mentioned above we have softened this idea by saying “contributed to” rather than “explaining”, and specified that this would only apply to CI-inducing *Wolbachia*.

Line 28: This is a bit misleading as many Wolbachia strains do not appear to be reproductive manipulators. Wolbachia's success is likely due to many things, including a high affinity for germline.

We thank the reviewer for pointing this out and have now reworded this section to specify that *Wolbachia* are successful insect colonizers, and some strains are also remarkable for their ability to manipulate reproduction, without necessarily linking the two.

Line 34: It would be useful to add a few words for non-experts, explaining the different names for CI genes (cif, cid, cin etc...). (Some of these different names are introduced later, but it would be much clearer to briefly explain this up front.)

We have added brief descriptions of the different nomenclature for *cif* vs *cid*.

Line 36: 'disputed' is a bit strong

We have changed this word to “debated”.

Line 42: It would be useful to explain to non-experts that wMel is a strain from Drosophila melanogaster flies that does not cause strong CI in its native host.

We agree this detail is useful, but believe it is more relevant to mention later in the discussion, so we include it there instead.

Line 69: It would be useful to explain to non-experts that wPip is a strain from Culex pipiens mosquitoes that causes strong CI in its native host.

We have added these details.

Line 149: It would be useful to explain to non-experts that wRec is a strain from Drosophila recens flies that causes strong CI in its native host.

We have added a short note of this nature.

Reviewer #4 (Remarks to the Author):

Wolbachia is an extraordinarily ecologically symbiont that infects about half of arthropod species, and this success is thought to be in a large part due to its ability to induce cytoplasmic incompatibility. This has been studied for over 50 years, but there has been a resurgence of interest in recent years as Wolbachia has been deployed to prevent mosquitoes transmitting dengue virus. The recent discovery of the genes underlying this trait was a major milestone on this field, and new insights in this area therefore have the potential to be of broad interest.

The headline result here is cifB expression in males is sufficient to induce cytoplasmic incompatibility. This is in line with classical models of the trait, but is different from Drosophila where there is strong evidence for cifA also being required (this protein 'rescues' the trait in females). This is an important insight. The second main result is that you can induce CI in Anopheles mosquitoes, but this is sensitive to the expression level of the factors involved. This will be important both for any application of Wolbachia in Anopheles to combat malaria, and also for understanding what might shape Wolbachia transfers between species in nature. A weakness of the paper is that there are no new insights into the molecular mechanisms of these effects, but in my judgement the results stand in the absence of this. The manuscript is clearly written and the results are straightforward. I have some minor suggestions.

We are grateful to the reviewer for their positive considerations of our work.

The observation that cifB is likely toxic but this toxicity is reduced by cifA is important, but the results supporting this were sometimes rather anecdotal. Line 72. 'but none that expressed cifB only, a suggestion that cifB may cause embryonic toxicity alleviated by cifA co-expression'. The same result in line 94 is repeated. There is no data shown for this in the results making this a weak way to infer lethality. I suggest this is deleted, or failing that it is presented as a properly

analysed experiment. The same conclusion about cifB lethality is mentioned later when cifB males are generated. Is it not straightforward to present some numbers to support the conclusion here?

We agree with the reviewer that in the original format the data presented was quite anecdotal and have removed these comments from the results. Additionally, we have added some data showing the number of embryos injected in each experiment in the methods section.

Line 62. It is important to cite a critique of the presence of Wolbachia in many species (Chrostek mBio) to acknowledge that some in the field are sceptical of these results. On the other hand, the absence of a citation for the preprint ‘Genomic and microscopic evidence of stable high density and maternally inherited Wolbachia infections in Anopheles mosquitoes’ is odd at this point, as this clearly shows Wolbachia does exist in some species at meaningful levels (I note this is cited later in a different context).

We now mention the detection of *Wolbachia* in *Anopheles* has been criticized for its robustness (Chrostek et al. 2019), although we personally believe that the higher titer *Wolbachia* infections in *An. moucheti* and *An. demeilloni* (now published) by Walker et al. 2021 support previous findings showing that *Wolbachia* can infect anophelines. We now cite Walker et al. in the introduction in addition to the discussion.

Line 20 “We report that CI can be fully recapitulated in these mosquitoes, and that cifB is sufficient to cause this reproductive manipulation”. This is confusing as the reproductive manipulation implies CI, which includes rescue, and therefore needs cifA. Reword.

Throughout the manuscript we have addressed this when relevant, specifying that we intend to describe the embryonic lethality associated with CI.

Line 101. Delete the subjective word ‘comparable’ (I look at this graph and see ‘less than’). Instead just state the mean hatch rate or something.

We have removed the word ‘comparable’ and described the data more objectively.

Figure 3C and the conclusions following from it. This assumes the two promoters drive expression in precisely the same cell types and stages of germline development. Is there evidence for this?

Subtle differences between expression patterns of *vasa* and *zpg* exist in flies, and we have now included this information when we introduce the *vasa* promoter. We understand the reviewer’s concerns that differences beyond expression level could contribute to the different induction of infertility between the *zpg-cifA; zpg-cifB* and *vasa-cifA; zpg-cifB* males. We have now elaborated on other possibilities for this different rescue within the manuscript, and changed the beginning of the results section to reflect the possibility that differences in timing or localization of *zpg* and *vasa*

promoter activity may be contributing to the different infertility phenotypes. Due to the different presentation of these results Fig. 3c and Fig. 3d have been swapped.

The Data Availability statement is not acceptable. It does not say where this data will be deposited. Sequence and annotation of constructs should be provided.

We apology for this erroneous omission, which we now have corrected. The sequencing and annotation of constructs will be certainly made available and accessible, and we have specified that this information will be available on GenBank

Decision Letter,third revision:

Dear Dr. Catteruccia,

Thank you for submitting your revised manuscript "Wolbachia cifB induces cytoplasmic incompatibility in the malaria mosquito" (NMICROBIOL-21030681C-Z). It has now been seen by the original referees and their comments are below. The reviewers find that the paper has improved in revision, and therefore we'll be happy in principle to publish it in Nature Microbiology, pending minor revisions to satisfy the referees' final requests and to comply with our editorial and formatting guidelines.

Thank you again for your interest in Nature Microbiology Please do not hesitate to contact me if you have any questions.

{redacted}

Reviewer #3 (Remarks to the Author):

I previously reviewed this manuscript. All of my concerns and suggestions have been addressed; the discussion and treatment of the literature on cif gene models and data, and on CI dynamics and spread, has been greatly strengthened. In my opinion, this paper is a useful and interesting contribution to the field of cytoplasmic incompatibility research, as it provides important insights into models and mechanisms of CI, and in a non-Drosophila system. The requirement of both cifB ('toxin?') and cifA ('rescue/modulating factor?') genes to render males incompatible may be particular to certain systems/strains. Beyond elucidating the mechanism of incompatibility, transgenic CI work in Anopheles also opens new possibilities for using CI approaches to control malaria.

Minor editorial notes:

- Line 73: Citation # 30 is now out in Current Biology
- Line 75: "When investigated, there was no evidence for CI in these strains.": This is a bit misleading since the strains reported in the Current Biology paper (#30) have some intact cif genes, as also pointed out later in this ms.
- Line 105: 'these' to 'this'
- Line 189: 'act' to 'acts'

Reviewer #4 (Remarks to the Author):

I have read the manuscript and am happy that all of my original comments (Reviewer 4) have been addressed in a careful and constructive way. This is an interesting set of results and I have no further comments to make.

Francis Jiggins

Decision Letter, final checks:

Dear Dr. Catteruccia,

Thank you for your patience as we've prepared the guidelines for final submission of your Nature Microbiology manuscript, "Wolbachia cifB induces cytoplasmic incompatibility in the malaria mosquito" (NMICROBIOL-21030681C-Z). Please carefully follow the step-by-step instructions provided in the attached file, and add a response in each row of the table to indicate the changes that you have made. Please also check and comment on any additional marked-up edits we have proposed within the text. Ensuring that each point is addressed will help to ensure that your revised manuscript can be swiftly handed over to our production team.

In recognition of the time and expertise our reviewers provide to Nature Microbiology's editorial process, we would like to formally acknowledge their contribution to the external peer review of your manuscript entitled "Wolbachia cifB induces cytoplasmic incompatibility in the malaria mosquito". For those reviewers who give their assent, we will be publishing their names alongside the published article.

Nature Microbiology offers a Transparent Peer Review option for new original research manuscripts submitted after December 1st, 2019. As part of this initiative, we encourage our authors to support increased transparency into the peer review process by agreeing to have the reviewer comments, author rebuttal letters, and editorial decision letters published as a Supplementary item. When you submit your final files please clearly state in your cover letter whether or not you would like to participate in this initiative. Please note that failure to state your preference will result in delays in accepting your manuscript for publication.

Cover suggestions

As you prepare your final files we encourage you to consider whether you have any images or illustrations that may be appropriate for use on the cover of Nature Microbiology.

Nature Microbiology has now transitioned to a unified Rights Collection system which will allow our Author Services team to quickly and easily collect the rights and permissions required to publish your work. Approximately 10 days after your paper is formally accepted, you will receive an email in providing you with a link to complete the grant of rights. If your paper is eligible for Open Access, our Author Services team will also be in touch regarding any additional information that may be required to arrange payment for your article.

Please note that *Nature Microbiology* is a Transformative Journal (TJ). Authors may publish their research with us through the traditional subscription access route or make their paper immediately open access through payment of an article-processing charge (APC). Authors will not be required to make a final decision about access to their article until it has been accepted. [Find out more about Transformative Journals](https://www.springernature.com/gp/open-research/transformative-journals)

Authors may need to take specific actions to achieve [compliance](https://www.springernature.com/gp/open-research/funding/policy-compliance-faqs) with funder and institutional open access mandates. For submissions from

January 2021, if your research is supported by a funder that requires immediate open access (e.g. according to [Plan S principles](https://www.springernature.com/gp/open-research/plan-s-compliance)) then you should select the gold OA route, and we will direct you to the compliant route where possible. For authors selecting the subscription publication route our standard licensing terms will need to be accepted, including our [self-archiving policies](https://www.springernature.com/gp/open-research/policies/journal-policies). Those standard licensing terms will supersede any other terms that the author or any third party may assert apply to any version of the manuscript.

{redacted}

{redacted}

Reviewer #3:

Remarks to the Author:

I previously reviewed this manuscript. All of my concerns and suggestions have been addressed; the discussion and treatment of the literature on cif gene models and data, and on CI dynamics and spread, has been greatly strengthened. In my opinion, this paper is a useful and interesting contribution to the field of cytoplasmic incompatibility research, as it provides important insights into models and mechanisms of CI, and in a non-Drosophila system. The requirement of both cifB ('toxin?') and cifA ('rescue/modulating factor?') genes to render males incompatible may be particular to certain systems/strains. Beyond elucidating the mechanism of incompatibility, transgenic CI work in Anopheles also opens new possibilities for using CI approaches to control malaria.

Minor editorial notes:

-Line 73: Citation # 30 is now out in Current Biology

-Line 75: "When investigated, there was no evidence for CI in these strains.": This is a bit misleading since the strains reported in the Current Biology paper (#30) have some intact cif genes, as also pointed out later in this ms.

-Line 105: 'these' to 'this'

-Line 189: 'act' to 'acts'

Reviewer #4:

Remarks to the Author:

I have read the manuscript and am happy that all of my original comments (Reviewer 4) have been addressed in a careful and constructive way. This is an interesting set of results and I have no further comments to make.

Francis Jiggins

Final Decision Letter:

Dear Professor Catteruccia,

I am pleased to accept your Article "Wolbachia cifB induces cytoplasmic incompatibility in the malaria mosquito vector" for publication in Nature Microbiology. Thank you for having chosen to submit your work to us and for working with me to refocus this paper appropriately for our readership.

Before your manuscript is typeset, we will edit the text to ensure it is intelligible to our wide readership and conforms to house style. We look particularly carefully at the titles of all papers to ensure that they are relatively brief and understandable.

Acceptance of your manuscript is conditional on all authors' agreement with our publication policies (see www.nature.com/nmicrobiolate/authors/gta/content-type/index.html). In particular your manuscript must not be published elsewhere and there must be no announcement of the work to any media outlet until the publication date (the day on which it is uploaded onto our website).

Please note that *Nature Microbiology* is a Transformative Journal (TJ). Authors may publish their research with us through the traditional subscription access route or make their paper immediately open access through payment of an article-processing charge (APC). Authors will not be required to make a final decision about access to their article until it has been accepted. [Find out more about Transformative Journals](https://www.springernature.com/gp/open-research/transformative-journals)

Authors may need to take specific actions to achieve [compliance](https://www.springernature.com/gp/open-research/funding/policy-compliance-faqs) with funder and institutional open access mandates. For submissions from January 2021, if your research is supported by a funder that requires immediate open access (e.g. according to [Plan S principles](https://www.springernature.com/gp/open-research/plan-s-compliance)) then you should select the gold OA route, and we will direct you to the compliant route where possible. For authors selecting the subscription publication route our standard licensing

terms will need to be accepted, including our [self-archiving policies](https://www.springernature.com/gp/open-research/policies/journal-policies). Those standard licensing terms will supersede any other terms that the author or any third party may assert apply to any version of the manuscript.
